# LONG-DOCUMENT QA WITH CHAIN-OF-STRUCTURED-THOUGHT AND FINE-TUNED SLMS

**Zhuowen Liang[1], Xiaotian Lin[1], Zhengxuan Zhang[1], Yuyu Luo[1], Haixun Wang[2], Nan Tang[1]***

[1]The Hong Kong University of Science and Technology (Guangzhou), [2]EvenUp, USA

## ABSTRACT

Large language models (LLMs) are widely applied to data analytics over documents, yet direct reasoning over long, noisy documents remains brittle and error-prone. Hence, we study document question answering (QA) that consolidates dispersed evidence into a structured output (*e.g.,* a table, graph, or chunks) to support reliable, verifiable QA. We propose a two-pillar framework, **LITECOST**, to achieve both high accuracy and low latency with small language models (SLMs). **Pillar 1: Chain-of-Structured-Thought (CoST).** We introduce a CoST template, a schema-aware instruction that guides a strong LLM to produce both a step-wise CoST trace and the corresponding structured output. The process induces a minimal structure, normalizes entities/units, aligns records, serializes the output, and verifies/refines it, yielding auditable supervision. **Pillar 2: SLM fine-tuning.** The compact models are trained on LLM-generated CoST data in two stages: Supervised Fine-Tuning for structural alignment, followed by Group Relative Policy Optimization (GRPO) incorporating triple rewards for answer/format quality and process consistency. By distilling structure-first behavior into SLMs, this approach achieves LLM-comparable quality on multi-domain long-document QA using 3B/7B SLMs, while delivering 2–4$\times$ lower latency than GPT-4o and DeepSeek-R1 (671B). The code is available at `https://github.com/HKUSTDial/LiteCoST`.

## 1 INTRODUCTION

Large language models (LLMs) are increasingly used for analytics (Chen et al., 2023; Zhu et al., 2025), yet direct reasoning over long documents is brittle and opaque, prone to errors in high-stakes domains such as finance and legal (Chew et al., 2023; Qin et al., 2024; Edge et al., 2024; Tang et al., 2024a). We therefore study long-document QA where explicit structured data helps. In this regime, the system constructs a query-specific structured data—*e.g.,* a table, graph, or chunks—from which final answer is directly derivable with explicit explanations. As shown in Fig. 1, extracting structured data for long-document QA (Edge et al., 2024; Zhang et al., 2025; Li et al., 2025c) improves reliability, interpretability, and reuse by exposing evidence and enabling routine verification.

We instantiate a query-conditioned pipeline: given a natural question $Q$ and documents $D$, the system induces a minimal schema tailored to $Q$, populates it with normalized evidence (*e.g.,* units, entities, time), and serializes it into a structured output $S$. The answer $A$ is then computed from $S$. Unlike fixed and pre-defined schemas, structures are assembled dynamically for each query, thereby excluding open-ended narrative questions that are not amenable to structured representations.

A natural idea is to directly leverage powerful LLMs (*e.g.,* GPT-4 or DeepSeek-R1) to emit the structured artifact. However, *direct prompting is not ideal*: (1) evidence is dispersed across long, multi-document contexts, leading to omissions or hallucinations; (2) values appear in heterogeneous units and formats, requiring normalization; and (3) long-context reasoning must remain consistent across the entire structure. As shown in Fig. 2(a), direct prompting often yields brittle results—omissions, hallucinations, and format drift (Wei et al., 2023; Wang et al., 2025). In contrast, Fig. 2(b) illustrates our **CoST template**, which guides the LLM to produce both (i) a schema-aligned **CoST trace** and (ii) a query-specific **serialized structured output (SSO)** (*e.g.,* table/graph/chunks), ensuring field completeness and format consistency, for robust and interpretable analytics over long-document QA.

---

*Corresponding author: Nan Tang (E-mail: nantang@hkust-gz.edu.cn)

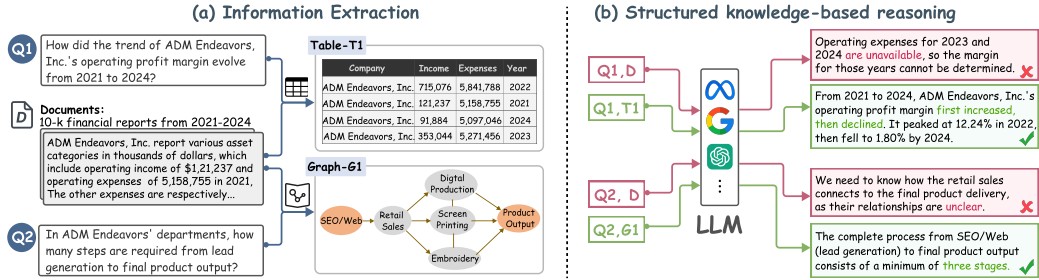

Figure 1: Structured data makes QA more accurate and reliable. (a) From raw document $D$, we extract a table $T_1$ for query $Q_1$ and a graph $G_1$ for query $Q_2$. (b) LLMs often fail when reasoning directly over unstructured text $(Q_1, D; Q_2, D)$, but succeed with structured inputs $(Q_1, T_1; Q_2, G_1)$.

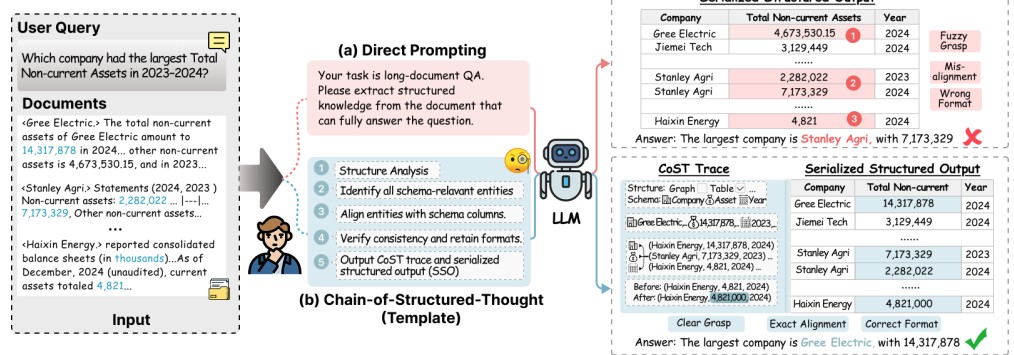

Figure 2: (a) Direct prompting LLMs often causes hallucinations and format errors. (b) (Question, Document, CoST Template) $\Rightarrow$ LLM $\Rightarrow$ (CoST Trace, SSO), yielding verifiable and auditable QA.

While CoST prompts with strong LLMs can yield accurate, verifiable SSOs, this effectiveness comes with a substantial cost: repeated large-model calls increase token/compute budgets, add latency, and limit throughput—undesirable for practical deployments that require low-latency, high-throughput service (Li et al., 2024; Xu et al., 2024). Reliance on hosted LLM APIs can also introduce privacy concerns for sensitive data. A natural response is to adopt small language models (SLMs)[1] for on-premises (on-prem) deployment, enabling cost-efficient inference in local or private environments; however, off-the-shelf SLMs struggle with the very skills CoST demands, including schema-aware extraction across long contexts, unit/entity normalization, record alignment, and step-consistent serialization, making naive LLM→SLM substitution ineffective (Tang et al., 2024b; Li et al., 2025b;a).

To balance *effectiveness* with *efficiency*, we introduce **LITECOST**, a two-pillar framework that equips SLMs with strong QA-by-structuring capabilities. **Pillar 1** invokes a powerful LLM once as a *structure-first trace generator*: it proposes a concise, query-conditioned schema and produces an auditable CoST trace together with structured data that makes evidence and formats explicit. **Pillar 2** *transfers* this ability to an SLM via a lightweight adaptation pipeline: supervised fine-tuning (SFT) to instill structure, format, and step discipline, followed by group-relative policy optimization (GRPO) that jointly rewards answer quality and process consistency.

**Contributions.** We propose LITECOST with three notable contributions:

1. **CoST for QA-by-Structuring:** a structure-first prompting paradigm that leverages LLMs to elicit step-wise, schema-guided CoST traces and SSOs from long, noisy documents—yielding auditable supervision and machine-checkable outputs.
2. **SLM adaptation via structured dual signals:** a two-phase SFT→GRPO recipe that introduces a novel dual-level reward, encompassing structured output quality as well as process consistency, to instill CoST-style, schema-aware structured reasoning into compact models.
3. **Empirical validation:** We have evaluated our approach across multiple domains, including finance, legal, and scientific literature. On the *financial* subset of the Loong benchmark (Wang et al., 2024), our CoST→SLM recipe substantially improves small models: LLaMA-3B gains **+27.6 accuracy points** and **+0.29 perfect rate (*PR*)**, while Qwen-7B gains **+17.8 accuracy** and **+0.22 *PR***, with the 7B model slightly surpassing GPT-4o. Inference is *2–4× **faster*** than

---

[1] We use *small language models* (SLMs) to denote compact models (*e.g.,* 3B–7B).

GPT-4o/DeepSeek-R1. Extensive experiments further show that our LITECOST framework delivers significant improvements while showcasing excellent generalization capabilities.

## 2 PRELIMINARY AND PROBLEM FORMULATION

**Long-Document QA as *QA-by-Structuring*.** We study a practically important regime of long-document QA in which the system turns a question $Q$ and a collection of long, noisy, multi-source documents $D$ into a compact, query-specific *serialized structured output* (SSO) with provenance. Structure becomes the interface: the system first induces a minimal schema tailored to $Q$, populates it with normalized, aligned evidence, and then derives the final answer from the structure with explicit support. This framing mitigates noise and dispersion, improves interpretability and reuse, and foregrounds four desiderata: *accuracy* (correct answers), *faithfulness* (evidence-grounded), *auditability* (verifiable traces), and *efficiency* (bounded compute/token cost).

**Chain-of-Structured-Thought (CoST).** In our design, the ***CoST template*** is the *input* to a language model: a schema-aware instruction that specifies step-wise, structure-first requirements. When executed, the language model produces two complementary *outputs*: (i) a ***CoST trace***, *i.e.,* the auditable, step-wise reasoning record that documents schema selection, evidence alignment, normalization, and verification; and (ii) a ***serialized structured output*** (SSO), *i.e.,* the machine-checkable artifact (table, graph, list, or record set) linked with provenance to the source documents. This input–output separation ensures that the LLM's role is well-defined: given a CoST template, it must emit both a reasoning trace and a structured output, enabling supervision, verification, and reuse. The whole **CoST** procedure consists of four key steps: (A1) structure analysis, (A2) trace generation, (A3) quality verification, and (A4) iterative refinement (see Sec. 3.1 for more details).

**Two Research Goals.** Our research goals are as follows.

(G1) *Accurate and verifiable QA*. Obtain *high-quality CoST traces and SSOs—i.e.,* schema-complete, format-consistent, and provenance-grounded outputs—from which we can compute correct answers.

(G2) *Low latency via SLMs.* Achieve *CoST-style reasoning at SLM speeds*. While strong LLMs are effective CoST generators, their latency/cost hinder deployment. Our objective is to transfer this structure-first behavior to compact models (SLMs) through fine-tuning, so that SLM-generated structures $S_{\text{SLM}}$ are as useful for answering as their LLM counterparts $S_{\text{LLM}}$, at much lower latency:

$$\text{LLM}(Q, S_{\text{SLM}}) \approx \text{LLM}(Q, S_{\text{LLM}}) \quad \text{and} \quad \text{Latency}(S_{\text{SLM}}) \ll \text{Latency}(S_{\text{LLM}}). \tag{1}$$

Operationally, we will (i) use LLMs once to generate high-quality CoST traces/SSOs (Pillar 1) and (ii) *fine-tune SLMs* to internalize schema/format/step discipline and process consistency (Pillar 2), enabling accurate, auditable QA at low cost.

## 3 LITECOST: FROM LLM COST GENERATION TO SLM ADAPTATION

Next, we present LITECOST (see Fig. 3), a two-stage framework designed to achieve the dual goals in Sec. 2: (G1) accurate and verifiable QA through high-quality CoST traces and SSOs, and (G2) low-latency execution via compact SLMs. In **Stage A**, a strong LLM executes the CoST template as input and produces auditable *CoST traces* and machine-checkable *SSOs* as outputs. These outputs serve as supervision signals that capture schema, normalization, alignment, and verification. In **Stage B**, we *transfer* this structure-first reasoning behavior into an SLM through a lightweight two-phase recipe: supervised fine-tuning (SFT) for schema/format/step compliance, followed by group-relative policy optimization (GRPO) to jointly reward answer quality and process consistency.

### 3.1 STAGE A (G1): COST (STRUCTURE-FIRST REASONING AND TRACE GENERATION)

As illustrated in Fig. 3, we operationalize CoST as a *structure-first, input→output* procedure: given a question, document, the ground truth answer, and CoST template, a strong LLM generates two outputs—an auditable *CoST trace* and a serialized structured output *SSO*.

**(A1) Structure Analysis.** The first step is to dynamically select the most suitable data structure and instantiate an accurate schema to support answering a given question. Specifically, LITECOST incorporates a question-oriented structure selection mechanism that, for example, chooses tables for statistical comparison or graphs for relational reasoning, without exhaustively processing the entire

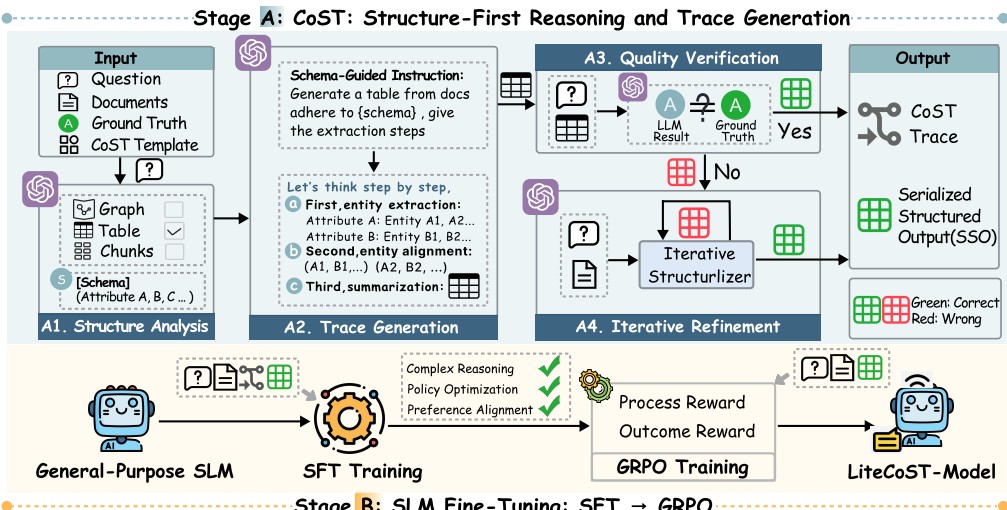

Figure 3: Overview of LITECOST, containing two stages: (1) CoST: Structure-First Reasoning and Trace Generation through structure analysis, trace generation, quality verification, and iterative refinement; and (2) SLM Fine-Tuning: SFT → GRPO process, including SFT for structure/format/steps, followed by GRPO with dual signals for answer/format quality and process consistency.

corpus. Once the structure type is chosen, we invoke a dynamic **schema construction** procedure in which the LLM parses the question and enumerates task-specific attributes/entities (*e.g.,* Company, Asset, Year), ensuring precise alignment with the question semantics.

**(A2) CoST Trace Generation.** Following structure analysis, we adopt an instruction-based chain-of-thought paradigm that performs step-by-step reasoning to progressively generate the trace to guide schema-aligned extraction. The task is specified through three key components: 1) **task description**, a template specifying step-wise requirements; 2) **input text**, the source documents; and 3) the dynamically generated **schema**. Guided by schema-informed instructions, a strong LLM extracts, aligns, and serializes into a deterministic structured format, emitting both the reasoning trace and the final structured output. The template of trace generation is provided in Appendix A.2.

**(A3) Quality Verification.** The module aims to assess the quality of the generated structured data by evaluating its ability to answer the original question. Since ground-truth structured data is unavailable, we adopt an LLM-as-Judge approach (Zheng et al., 2023), where a strong LLM evaluator (e.g., GPT-4o) assesses the extracted responses. Inference outputs that exactly match the reference answers are deemed correct and retained for subsequent training. Further details are provided in Appendix A.3.

**(A4) Iterative Refinement.** At its core, the module employs an **Iterative Structuralizer** that refines low-quality samples by regenerating structured knowledge for GRPO training. Rather than discarding flawed but challenging cases, it reuses them recursively with the question and context, reframing the task as supplemental extraction and providing richer supervision than vanilla fine-tuning. The iterative update rule, sufficiency evaluator, and stopping criteria are detailed in Appendix A.4.

**Final Output.** After the CoST pipeline, the final output is $(c^*, S^*)$, where $c^*$ is the CoST trace (generated in A2), and $S^*$ the structured output refined through quality verification (A3) and iterative refinement (A4). This pair provides high-quality supervision for training and downstream reasoning.

## 3.2  STAGE B (G2): SLM FINE-TUNING (SFT → GRPO)

Given the supervised training data, LITECOST first warms up the model with Supervised Fine-Tuning (SFT). We then apply reinforcement learning with Group Relative Policy Optimization (GRPO), introducing a dual-level reward that jointly optimizes (1) outcome reward, which evaluates the format compliance and answer correctness, and (2) process reward, which scores step-wise reasoning against ground-truth evidence to enforce a reliable extraction path.

**Training Data Template.** Each training sample is defined as $z = (i, d, c^*, y^*)$, where $i$ is the question, $d$ the document input, $c^*$ the CoST reasoning trace (enclosed by <reasoning>...</reasoning>), and $y^*$ the structured output (enclosed by <answer>...</answer>). In *SFT*, the model learns to generate $(c^*, T^*)$ from $(i, d)$, while *GRPO* also conditions on $(i, d)$ and optimizes with dual-level rewards (process, outcome) against the verified targets.

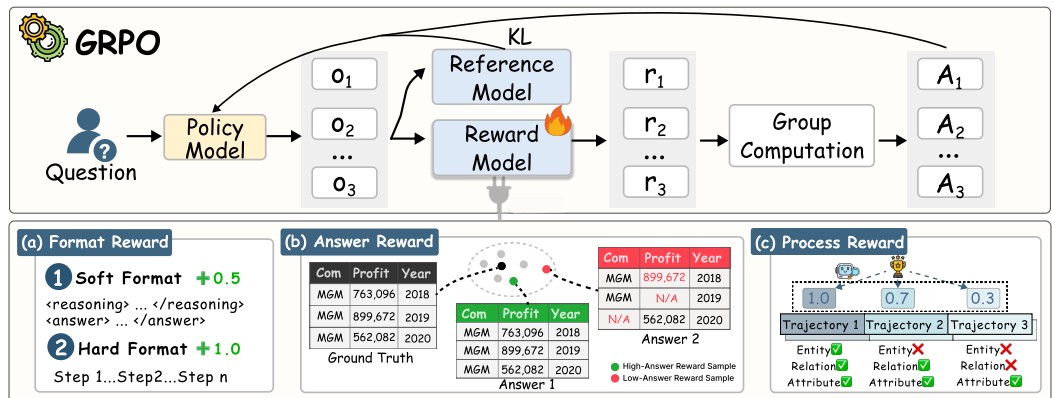

Figure 4: The GRPO training pipeline based on dual-level reward.

**Supervised Fine-tuning (SFT).** We initially performed Supervised Fine-Tuning (SFT) on a general-purpose base model, specifically enhancing its capability for CoT-driven information extraction. This process enables the model to acquire fundamental extraction capability (*e.g.,* handling structure, format, and step-wise reasoning) in a specific domain, thus substantially mitigating the errors observed when deploying the base model on complex extraction tasks.

**Group Relative Policy Optimization (GRPO).** We then employ GRPO via a three-level reward mechanism, as illustrated in Fig. 4.

*Formulation.* For each question $q$, GRPO samples a group of outputs $\{o_1, o_2, \ldots, o_G\}$ from the old policy $\pi_\theta$. Each output receives a reward $r_i$, yielding a set of $G$ rewards $\mathbf{r} = \{r_1, r_2, \ldots, r_G\}$. From these rewards, we compute the group-relative advantage $A_i = \frac{r_i - \text{mean}(r_1 \ldots r_G)}{\text{std}(r_1 \ldots r_G)}$ and then optimizes the policy model by maximizing the following objective:

$$\mathcal{J}_{\text{GRPO}}(\theta) = \mathbb{E}\left[\mathbf{v} \sim P(\mathbf{V}), \{o_i\}_{i=1}^G \sim \pi_{\theta_{\text{old}}}(\mathcal{O}|\mathbf{v})\right]$$
$$\frac{1}{G}\sum_{i=1}^G \left(\min\left(r_i^{\text{ratio}} A_i, \text{clip}\left(r_i^{\text{ratio}}, 1-\epsilon, 1+\epsilon\right) A_i\right) - \beta D_{\text{KL}}(\pi_\theta \| \pi_{\text{ref}})\right), \tag{2}$$

where $\beta$ and $\epsilon$ are hyper-parameters, $r_i^{\text{ratio}}$ is the importance sampling ratio comparing the likelihood of output $o_i$ under the new and old policies, and $A_i$ is the group-relative advantage. The clipping operator would stabilize updates within a trust region, and the minimum operation ensures conservative yet effective policy updates (Shao et al., 2024). Further details are provided in Appendix B.1.

*Format Compliance.* Fig. 4 (a) illustrates the hierarchical design of the format reward: a soft reward (0.5) is given if the output contains a single reasoning sequence in <reasoning> and a final answer in <answer> without extraneous content; a hard reward (1.0) is assigned if the reasoning is further structured with explicit step labels (*e.g.,* Step 1, Step 2); otherwise, the score is 0.

*Answer Correctness.* As illustrated in Fig. 4(b), we address the limitations of rule-based evaluation by adopting a hybrid metric that combines structural alignment and semantic similarity:

$$f_{\text{score}} = \alpha \cdot \mathcal{S}_{\text{struct}} + (1-\alpha) \cdot \mathcal{S}_{\text{sem}} \tag{3}$$

For $\mathcal{S}_{\text{struct}}$, we use rule-based checks (*e.g.,* row-column alignment in tables) to verify structural correctness. For $\mathcal{S}_{\text{sem}}$, we adopt GPT-4o-mini as an automatic evaluator, comparing the content within <answer>...</answer> tags against the reference; outputs with higher semantic similarity receive higher rewards. The raw score $f_{\text{score}}$ is scaled from $[0, 100]$ to $[0, 1]$, with NULL rewards assigned to empty outputs. The detailed LLM-based evaluation prompts are provided in Appendix B.2.

*Process Reward.* Outcome rewards alone are sparse and insufficient for fine-grained guidance. We therefore introduce a consistency-based process reward to supervise reasoning at the step level. Consistency is evaluated from both the entity-level and the tuple-level, enabling the model to capture fine-grained errors such as partially incorrect entities or mismatched relations. For each step $i$, LLM is prompted with an instruction $I_{\text{consistency}}$ to judge whether the predicted step result $s_i$ is consistent with the corresponding ground truth $s_i^*$. If the consistency holds, the step is assigned a score of 1; otherwise, it receives 0. The overall process reward is formally defined as:

$$R_{\text{process}}(s_i) = \frac{1}{N}\sum_{i=1}^N \mathbf{1}\left[\text{Cons}(s_i, s_i^* \mid I_{\text{consistency}})\right], \tag{4}$$

where $N$ denotes the total number of reasoning steps, $\text{Cons}(\cdot)$ is the LLM-based consistency function, and $\mathbf{1}[\cdot]$ is the indicator function that returns 1 if the consistency check is satisfied and 0 otherwise. This formulation provides a dense and fine-grained training signal, guiding the model towards faithful step-by-step extraction while complementing the sparse outcome reward, as shown in Fig. 4 (c).

*Overall Reward.* The overall reward is defined as the sum of the format compliance, answer correctness, and process rewards. To prevent training dynamics from being dominated by other reward signals, we introduce a scaling factor that modulates the process reward along each trajectory, $\tilde{R}_{\text{process}}(s_i) = R_{\text{process}}(s_i) \cdot \gamma(T_i)$. Here, $\gamma(T_i)$ is a trajectory-level coefficient: positive for correct answers to reinforce reasoning, negative for incorrect or overthought trajectories to discourage such behaviors, and 1 for format errors to isolate penalties to specific steps.

## 4 EXPERIMENTS ON LONG-DOCUMENT QA WITH COST AND SLMS

In this section, we evaluate the performance of our proposed LITECOST framework on the Loong benchmark (Wang et al., 2024), which effectively captures the challenges of generating *serialized structured output (SSO)* across varying context lengths. Rather than directly answering questions, we focus on the ability of LITECOST to produce reliable SSO that supports long-document QA. We further assess its efficiency and conduct ablation studies to analyze contributing factors. Specifically, we aim to address the following research questions:

**(1) Benefits of Structured Data:** How do structured outputs enhance long-document QA?
**(2) Effectiveness:** How effective is LITECOST in generating high-quality SSO for long-document QA, compared with current LLMs and state-of-the-art methods?
**(3) Efficiency:** How efficient is LITECOST relative to LLMs in terms of SSO generation speed?
**(4) Ablation Study:** What factors contribute to performance gains in structured output generation?
**(5) Generalization:** How well does the framework generalize to other datasets and domains?

### 4.1 EXPERIMENTAL SETUP

**Training Dataset.** To support two-phase training, we construct two domain-specific datasets via LITECOST from four large-scale multi-task resources: FINQA (Chen et al., 2021), TAT-QA (Zhu et al., 2021), SQUAD (Rajpurkar et al., 2016), and LEGALBENCH (Pipitone & Alami, 2024). The datasets target the *finance*, *legal* and *general* knowledge, capturing diverse reasoning patterns (e.g., aggregation, comparison, multi-hop inference) in realistic settings drawn from diverse documents.

**Evaluation Dataset.** We adopt the Loong benchmark (Wang et al., 2024), a real-world multi-document QA dataset with 1,600 test samples spanning three domains (Finance, Legal, Paper), four task categories (Spotlight Locating, Comparison, Clustering, Chain of Reasoning), and four document length settings where longer contexts disperse relevant information. Our analysis focuses primarily on an in-depth analysis of the finance domain, with legal results provided in Appendix E.1.

**Evaluation Details.** Defining ground truth for the structured output from long-context documents, poses significant challenges. To address this, we adopt a 2-hop evaluation paradigm by leveraging downstream QA tasks (Jain et al., 2024). Specifically, we employ the GPT-4o as an automatic judge, scoring model responses from 0 to 100 based on accuracy, hallucination, and completeness. It also introduces the Perfect Rate, which measures the proportion of responses that achieve a perfect score.

**Baselines.** To comprehensively evaluate the generation capability of LITECOST, we compare the performance gains achieved through both LLMs and SLMs. Specifically, we consider two categories of baselines: the first targets reasoning, where LLMs are prompted to generate structured output (*e.g.,* Zero-shot and Chain-of-Thought (CoT) (Wei et al., 2022)). The second focuses on improvements with SLMs, where we compare against several state-of-the-art models, including Llama3.2-3B-Instruct, Qwen2-7B-Instruct, Llama-3.1-8B-Instruct, Qwen2.5-14B-Instruct, GPT4o-mini, GPT-4o, and DeepSeek-R1. We further include two categories of baselines: (1) Fine-tuned IE models, such as ODIE (Jiao et al., 2023), IEpile (Gui et al., 2024b), and Struc-bench (Tang et al., 2024b); and (2) Modular extraction frameworks that leverage component modules to extract structured knowledge, including StructRAG (Li et al., 2025c). For a fair comparison, we evaluate the baseline methods using the same backbones (*i.e.,* LLaMA-3.2-3B-Instruct, Qwen2-7B-Instruct) as those used for our LITECOST. These baselines act as structured output generator, with GPT-4o employed as the reasoning model to produce responses. More details are provided in Appendix C.2.

Table 1: Comparison of different models generating structured outputs for long-document QA on the *Finance* Subset of Loong. Green highlights the remarkable improvements over the base model.

| Model | Model Size | Spotlight Locating | | Comparison | | Clustering | | Chain of Reasoning | | Overall | |
|---|---|---|---|---|---|---|---|---|---|---|---|
| | | AS | PR | AS | PR | AS | PR | AS | PR | AS | PR |
| *Close-Sourced Models & Large Language Models* | | | | | | | | | | | |
| LLaMA-3.1-8B-Instruct | 8B | 55.03 | 0.20 | 51.60 | 0.15 | 51.50 | 0.04 | 44.75 | 0.02 | 51.32 | 0.10 |
| GPT-4o-mini | 8B | 84.42 | 0.70 | 80.40 | 0.67 | 77.38 | 0.40 | 65.35 | 0.18 | 78.08 | 0.51 |
| Qwen2.5-14B-Instruct | 14B | 83.74 | 0.57 | 82.12 | 0.56 | 69.96 | 0.24 | 66.41 | 0.10 | 75.60 | 0.38 |
| GPT-4o (Abacha et al., 2025) | 200B | 84.10 | 0.73 | 80.53 | 0.60 | 81.50 | 0.50 | 64.30 | 0.25 | 79.32 | 0.54 |
| Deepseek-R1 (Guo et al., 2025) | 671B | 84.27 | 0.62 | 78.97 | 0.55 | 75.42 | 0.34 | 74.40 | 0.35 | 78.18 | 0.46 |
| LLaMA-3.2-3B-Instruct (Base) | 3B | 49.90 | 0.16 | 52.10 | 0.14 | 47.89 | 0.07 | 46.85 | 0.06 | 49.37 | 0.11 |
| LLaMA-3.2-3B-Instruct (*Ours*) | 3B | **81.27** | **0.53** | **78.08** | **0.49** | **78.34** | **0.36** | **64.75** | **0.16** | **76.95** | **0.40** |
| | | ↑31.37 | ↑0.37 | ↑25.98 | ↑0.35 | ↑30.45 | ↑0.29 | ↑17.90 | ↑0.10 | ↑27.58 | ↑0.29 |
| Qwen2-7B-Instruct (Base) | 7B | 63.10 | 0.36 | 67.85 | 0.37 | 60.83 | 0.18 | 52.25 | 0.09 | 62.10 | 0.26 |
| Qwen2-7B-Instruct (*Ours*) | 7B | **83.97** | **0.62** | **81.55** | **0.59** | **81.00** | **0.43** | **67.98** | **0.18** | **79.93** | **0.48** |
| | | ↑20.87 | ↑0.26 | ↑13.70 | ↑0.22 | ↑20.17 | ↑0.25 | ↑15.73 | ↑0.09 | ↑17.83 | ↑0.22 |

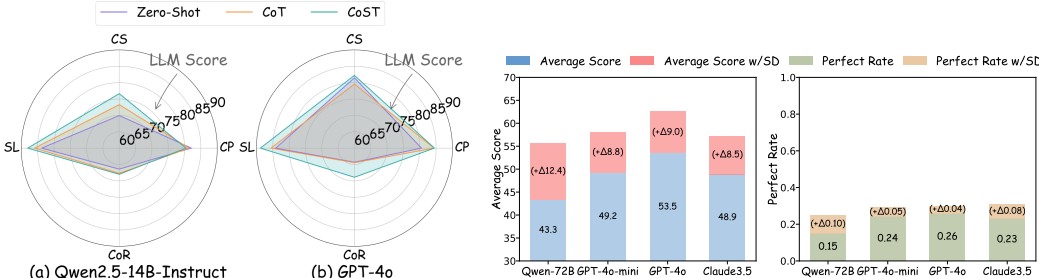

Figure 5: Radar plot of detailed scores for different prompting methods on 4 subtasks on the *Finance* subset of Loong.

Figure 6: Quality assessment of CoST-generated structured data via reasoning performance on Loong across popular LLMs (*SD* denotes structured data).

**Implementation Details.** During the training phase, LITECOST employs a two-phase training pipeline comprising LoRA fine-tuning followed by GRPO optimization using trl and verl. The model is trained in two stages: first, fine-tuning for 3 epochs with a learning rate of 2e-4 and batch size of 16 using a LoRA adapter (rank 16, lora alpha 32); followed by reinforcement learning with GRPO using a learning rate of 1e-5, batch size of 16, and 5 sampled generations per query. In Equation 3, the weighting parameter $\alpha$ is set to 0.3; the training cost is about \$20, and the maximum generation length is extended to 2,048 tokens to support CoT-style reasoning.

## 4.2 THE BENEFITS OF STRUCTURED DATA

This experiment evaluates how structured data improves model performance on knowledge-intensive reasoning, emphasizing the need for accurate serialized structured output (SSO). The data is curated by LITECOST using GPT-4o as the base model; structure distributions are detailed in Appendix C.5.

**High-quality SSO from CoST improves LLM Reasoning.** As shown in Fig. 6, all models achieve stronger reasoning when leveraging the structured data rather than raw long documents. With LITECOST, overall scores rise by 12.41, 8.77, 9.04, and 8.47 points, and perfect rates improve by +0.10, +0.05, +0.04, and +0.08 for Qwen2-72B-Instruct, GPT-4o-mini, GPT-4o, and Claude-3.5-Sonnet, respectively. These consistent gains highlight the value of high-quality structured knowledge in improving both accuracy and reliability. Full results are provided in Appendix C.5.

## 4.3 EFFECTIVENESS: HOW GOOD IS LITECOST FOR SSO GENERATION?

In this section, we evaluate the effectiveness of LITECOST across both LLMs and compact SLMs on the *Finance* subset of Loong, compared with state-of-the-art models and baseline methods. The in-depth analysis demonstrate that LITECOST consistently outperforms other comparable strategies, achieving substantial gains in correctness, and Sec. 4.6 further confirms its broader generalization.

**Efficacy of Chain-of-Structured-Thought.** Fig. 5 presents a radar chart comparing the performance of different prompting methods across four task categories, evaluated on two backbone LLMs. The results show that step-wise reasoning substantially improves SSO generation and yields high-quality

Table 2: Performance of the *Finance* subset of Loong compared with other state-of-the-art methods

| Backbone | Method | Spotlight Locating | | Comparison | | Clustering | | Chain of Reasoning | | Overall | |
|---|---|---|---|---|---|---|---|---|---|---|---|
| | | AS | PR | AS | PR | AS | PR | AS | PR | AS | PR |
| LLaMA-3.2-3B-Ins | ODIE (Jiao et al., 2023) | 68.89 | 0.39 | 61.30 | 0.30 | 61.11 | 0.15 | 49.75 | 0.07 | 61.21 | 0.23 |
| | IEpile (Gui et al., 2024b) | 62.90 | 0.37 | 65.10 | 0.30 | 63.12 | 0.14 | 50.95 | 0.02 | 61.90 | 0.22 |
| | Struc-bench (Tang et al., 2024b) | 55.13 | 0.15 | 51.05 | 0.15 | 48.16 | 0.06 | 44.10 | 0.08 | 49.90 | 0.11 |
| | StructRAG (Li et al., 2025c) | 39.50 | 0.01 | 39.70 | 0.02 | 31.08 | 0.00 | 35.95 | 0.00 | 36.04 | 0.01 |
| | LITECOST (Ours) | **81.27** | **0.53** | **78.08** | **0.49** | **78.34** | **0.36** | **64.75** | **0.16** | **76.95** | **0.40** |
| Qwen2-7B-Ins | ODIE (Jiao et al., 2023) | 82.13 | 0.59 | 73.85 | 0.48 | 73.54 | 0.32 | 55.30 | 0.12 | 72.86 | 0.40 |
| | IEpile (Gui et al., 2024b) | 71.83 | 0.45 | 72.60 | 0.46 | 70.86 | 0.29 | 54.20 | 0.11 | 69.19 | 0.35 |
| | Struc-bench (Tang et al., 2024b) | 81.60 | **0.63** | 74.90 | 0.53 | 73.40 | 0.36 | 60.35 | 0.19 | 73.72 | 0.44 |
| | StructRAG (Li et al., 2025c) | 48.83 | 0.03 | 46.80 | 0.02 | 55.34 | 0.06 | 42.55 | 0.00 | 49.68 | 0.03 |
| | LITECOST (Ours) | **83.97** | 0.62 | **81.55** | **0.59** | **81.00** | **0.43** | **67.98** | **0.18** | **79.93** | **0.48** |

supervision signals, with CoT consistently outperforming Zero-Shot, especially on Qwen2.5-14B-Instruct. Beyond this, our structured prompting paradigm (CoST) yields the strongest improvements, consistently reaching the outer boundary (green line) and achieving top performance across nearly all tasks, notably in Chain of Reasoning (GPT-4o), Clustering (Qwen2.5-14B-Ins), and Spotlight Locating, underscoring its consistent effectiveness. See full numerical results in the Appendix C.3.

**Our fine-tuned SLMs ≫ other SLMs.** Table 1 demonstrates that our model consistently outperforms all evaluated small language models (defined as open-source models with fewer than 8 billion parameters). Both fine-tuned variants, **LLaMA-LiteCoST** and **Qwen-LiteCoST**, achieve substantial improvements over their base models, with gains of (+27.58, +0.29) and (+17.83, +0.22), and consistent enhancements across all sub-tasks. Compared with other small scales, the LLaMA-LiteCoST significantly outperforms the 7B, and 8B models by (+14.85, +0.14) and (+25.63, +0.30) in overall score and perfect rate, respectively, while Qwen-LiteCoST delivers even larger improvements, underscoring the effectiveness of our approach.

**Our fine-tuned SLMs ≈ LLMs.** On one hand, both variants surpass Qwen-14B-Instruct despite having far fewer parameters, with improvements of (+1.35, +0.02) on the LLaMA backbone and (+4.33, +0.10) on the Qwen backbone. On the other hand, **Qwen-LiteCoST** achieves the best overall performance among all evaluated models, surpassing all small baselines and even exceeding GPT-4o-mini by 1.85, Deepseek-R1 by 1.75, and GPT-4o by 0.61. It achieves top-2 performance on 5 out of 8 evaluation points across the four subtasks, highlighting the effectiveness of our training strategy in narrowing the gap between lightweight and large-scale models for structured output generation. Full results across different document sizes are provided in Appendix F.

**RL-enhanced gains for SLMs.** Compared with other state-of-the-art methods, including fine-tuned models and modular extraction frameworks, LITECOST achieves superior performance across all tasks, as shown in Table 2. In particular, LITECOST substantially outperforms the strong baseline StructRAG, with gains of (+30.91, +0.39) on LLaMA and (+30.47, +0.46) on Qwen, Moreover, relative to the previous best fine-tuned methods, it sets a new state of the art, achieving improvements of (+15.05, +0.18) over IEPile on LLaMA and (+6.41, +0.05) over Strucbench on Qwen. These results indicate that our RL-enhanced framework provides a principled advancement over conventional fine-tuning for information extraction, further underscoring its effectiveness and robustness.

### 4.4 EFFICIENCY: HOW FAST ARE SLMs COMPARED WITH LLMs FOR SSO GENERATION?

**SLMs are much faster than LLMs.** The latency comparison in Fig. 7 demonstrates that our model offers an optimal trade-off between accuracy and efficiency in structured output generation tasks, measured as the average time per sample on the Loong dataset. **Qwen-LiteCoST** attains lower latency (12.09s) than LLaMA3.1-8B-Instruct (13.19s) and Qwen2.5-14B-Instruct (14.71s), while delivering substantial accuracy gains. Notably, it maintains latency comparable to its base model (Qwen2-7B-Instruct at 11.89s), while running nearly *2×* faster than GPT-4o (21.15s) and *4×* faster than DeepSeek-R1 (44.44s),

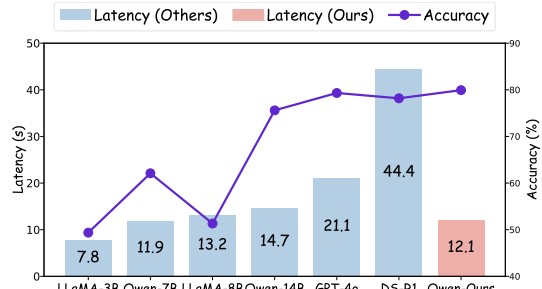

Figure 7: Comparison of extraction time and accuracy across different scale models.

without relying on proprietary APIs. For scenarios requiring faster extraction, LLaMA-LiteCoST is preferable, running in just 8.04s while achieving performance comparable to 14B-scale models.

Table 3: Effect of different reward designs in ablation study on the *Finance* subset of Loong.

| Model | Spotlight Locating | | Comparison | | Clustering | | Chain of Reasoning | | Overall | |
|---|---|---|---|---|---|---|---|---|---|---|
| | AS | PR | AS | PR | AS | PR | AS | PR | AS | PR |
| LLaMA-Ours | **81.27** | **0.53** | **78.08** | **0.49** | **78.34** | **0.36** | 64.75 | **0.16** | **76.95** | **0.40** |
| w/o Process Reward | 79.10 | 0.48 | 77.95 | 0.47 | 76.03 | 0.30 | 63.98 | 0.13 | 75.52 | 0.37 |
| w/o Outcome Reward | 76.07 | 0.45 | 73.10 | 0.41 | 71.72 | 0.23 | **68.22** | 0.15 | 72.55 | 0.32 |
| Qwen-Ours | 83.97 | **0.62** | **81.55** | **0.59** | **81.00** | **0.43** | **67.98** | **0.18** | **79.93** | **0.48** |
| w/o Process Reward | **87.93** | 0.63 | 77.95 | 0.55 | 77.32 | 0.42 | 60.60 | 0.15 | 77.39 | 0.46 |
| w/o Outcome Reward | 86.29 | 0.57 | 75.65 | 0.45 | 73.56 | 0.31 | 63.35 | 0.18 | 75.43 | 0.39 |

Table 4: Performance Comparison on the *Legal* subset of Loong. Green highlights the remarkable improvements over the base model, while Red indicates relative drops.

| IE Model | Model Size | Spotlight Locating | | Comparison | | Clustering | | Chain of Reasoning | | Overall | |
|---|---|---|---|---|---|---|---|---|---|---|---|
| | | AS | PR | AS | PR | AS | PR | AS | PR | AS | PR |
| *Close-Sourced Models & Large Language Models* | | | | | | | | | | | |
| GPT-4o-mini | 8B | 46.55 | 0.10 | 28.05 | 0.00 | 48.68 | 0.13 | 42.56 | 0.11 | 41.94 | 0.09 |
| Qwen2.5-14B-Instruct | 14B | 48.45 | 0.08 | 21.90 | 0.01 | 57.31 | 0.16 | 26.74 | 0.02 | 37.51 | 0.06 |
| GPT-4o | 200B | 50.05 | 0.06 | 27.00 | 0.01 | 61.16 | 0.14 | 33.11 | 0.03 | 42.06 | 0.06 |
| LLaMA-3.2-3B-Instruct (Base) | 3B | 41.00 | 0.08 | 25.10 | 0.01 | 31.74 | 0.02 | 27.29 | 0.01 | 30.67 | 0.03 |
| LLaMA-3.2-3B-Instruct (*Ours*) | 3B | **62.20** | **0.30** | **45.20** | **0.02** | **45.00** | **0.09** | **36.55** | **0.02** | **45.45** | **0.09** |
| | | ↑21.20 | ↑0.22 | ↑20.10 | ↑0.01 | ↑13.26 | ↑0.07 | ↑9.26 | ↑0.01 | ↑14.78 | ↑0.06 |
| Qwen2-7B-Instruct (Base) | 7B | 37.90 | 0.05 | 18.90 | 0.00 | 57.85 | 0.18 | 35.44 | 0.08 | 38.05 | 0.08 |
| Qwen2-7B-Instruct (*Ours*) | 7B | **52.85** | **0.12** | **31.00** | **0.00** | **60.37** | **0.22** | **37.88** | **0.06** | **44.94** | **0.10** |
| | | ↑14.95 | ↑0.07 | ↑12.10 | ↑0.00 | ↑2.52 | ↑0.04 | ↑2.44 | ↓0.02 | ↑6.89 | ↑0.02 |

## 4.5 ABLATION STUDY

**Effect of Different Reward.** To identify the key drivers of LITECOST 's performance, we ablated reinforcement learning configurations (Table 3). When we remove the process reward component from the complete model, performance drops by 1.43 points on the LLaMA backbone (76.95→75.52) and 2.54 points on Qwen (79.93→77.39). This demonstrates that fine-grained process rewards effectively guide step-wise extraction, complementing outcome-based signals to yield stronger overall performance. Similarly, excluding the outcome reward causes a much larger drop (–4.40 on LLaMA, -4.50 on Qwen), underscoring its importance for answer correctness. These results highlight the synergistic effects of process- and outcome-level supervision, yielding a more robust training strategy. Case studies in Appendix D further illustrate how RL-enhancement improves extraction quality.

## 4.6 GENERALIZATION

To further demonstrate the broad applicability of LITECOST beyond financial analysis, we extend our evaluation to two additional distinct domains: Legal and Scientific Question Answering.

**Legal Domain.** We evaluate performance on Loong legal subset (Wang et al., 2024) to assess the capability of LITECOST in generating serialized structured outputs within complex legal contexts. As shown in Table 4, our LITECOST-tuned 3B and 7B models substantially outperform their base versions, achieving Average Score and Perfect Rate gains of (+14.78, +0.06) and (+6.89, +0.02), respectively. Remarkably, both compact variants surpass significantly larger models despite having far fewer parameters. Specifically, the LLaMA-based model outperforms Qwen2.5-14B-Instruct, GPT-4o-mini, and GPT-4o by (+7.94, +0.03), (+3.51, +0.00), and (+3.39, +0.03), respectively, with the Qwen-based variant demonstrating similar superiority. More details are discussed in Appendix E.1.

**Results on Open-domain QA.** In addition to the Loong benchmark, we verify the performance of **CoST** and LITECOST on LongBench (Bai et al., 2024), a comprehensive benchmark tailored for multi-task long-document QA that covers key long-text application scenarios. Our analysis focuses on both single-document and multi-document QA tasks across four datasets. We compare LITECOST against state-of-the-art models using the standard F1 score, which assesses the quality of reasoning results derived from the extracted information. Consistent with the experimental setup in Loong, GPT-4o is employed as the reasoning agent to generate final answers based on the structured outputs.

As presented in Table 5, the results align with our findings in other domains, highlighting two consistent insights: (1) **CoST enhances LLM reasoning** (Table 5a): applying CoST consistently boosts performance for both Qwen2.5-14B-Instruct and GPT-4o across all datasets, yielding F1

Table 5: Performance comparison on LongBench benchmark

(a) Quality assessment of CoST-generated structured data via reasoning performance on LongBench across LLMs (*SD* denotes structured data).

| Model | Single-doc | | Multi-docs | |
|---|---|---|---|---|
| | **NarQA** | **Qasper** | **HotpotQA** | **2Wiki** |
| Qwen-14B | 29.78 | 45.07 | 62.59 | 60.00 |
| w/*SD* | **31.77** | **47.17** | **67.41** | **67.11** |
| GPT-4o | 32.59 | 46.80 | 70.93 | 67.75 |
| w/*SD* | **35.09** | **49.28** | **73.47** | **72.98** |

(b) Performance Comparison Between LiteCoST-Tuned Models and LLM-Based Baselines on LongBench.

| IE Model | Single-doc | | Multi-docs | |
|---|---|---|---|---|
| | **NarQA** | **Qasper** | **HotpotQA** | **2Wiki** |
| LLaMA-3.2-3B | 16.94 | 34.46 | 54.92 | 51.82 |
| Qwen2-7B | 19.49 | 35.67 | 45.06 | 41.40 |
| GPT-4o-mini | 24.38 | 40.28 | 65.03 | 65.15 |
| GPT-4o | 28.68 | 43.39 | 67.68 | **68.29** |
| LLaMA-LiteCoST | 27.24 | 41.37 | 66.86 | 67.52 |
| Qwen-LiteCoST | **30.40** | **44.64** | **68.39** | 65.73 |

gains of up to **+7.11** and **+5.23** points, respectively. (2) **SLMs rival proprietary models** (Table 5b): Qwen-LiteCoST achieves the best performance on NarrativeQA, Qasper, and HotpotQA, surpassing GPT-4o by **0.71–1.72** points. Notably, it improves over its base model by substantial margins (up to +23.33), with both LITECOST variants consistently ranking in the **top three** across all datasets. Collectively, these results demonstrate the strong effectiveness and broad generality of CoST and LITECOST across diverse domains and varying task complexities.

## 5 RELATED WORK

**Long-Document Question Answering** is a critical test of LLM reasoning (Wang et al., 2024; Zhang et al., 2024), where dispersed evidence, noise, and complex reasoning make it more challenging than short-passage QA. Existing approaches, including long-context models (Yang et al., 2024; Guo et al., 2025), retrieval augmentation (Lewis et al., 2020), and chain-of-thought prompting (Wei et al., 2022), mitigate these challenges but remain brittle, often yielding hallucinations in high-stakes domains. Structured knowledge has also been explored (Li et al., 2025c; Panda et al., 2024; Edge et al., 2024; Chen et al., 2023), yet such methods require repeated large-LLM calls, leading to high cost and limited scalability. To address this, we propose a structure-first design with efficient SLM execution.

**LLMs for Long-Context Information Extraction.** Information Extraction (IE) underpins many downstream NLP tasks (Xu et al., 2024; Zhang et al.). While large language models (LLMs) perform well on diverse IE tasks, even in zero- and few-shot settings (Lu et al., 2023; Wei et al., 2023; Ashok & Lipton, 2023; Wang et al., 2023b; 2025; Jain et al., 2024), existing methods remain confined to short texts. In long contexts, dispersed evidence and noise impede reliable integration. Prior "QA-by-structuring" systems (Li et al., 2025c; Tang et al., 2024b) focus on structured generation rather than verifiable step-wise extraction, and thus do not achieve reliable evidence tracing. To address this, we propose the *Chain-of-Structured-Thought (CoST)* paradigm, which uses step-wise reasoning for structured extraction, yielding schema-aligned outputs and rich supervision for fine-tuning.

**Fine-tuned Lightweight Models.** While LLMs provide high-quality extraction, their computational cost and latency limit real-time use. Fine-tuned smaller models improve efficiency (Gui et al., 2024a; Xiao et al., 2023; Wang et al., 2023a; Wu et al., 2025), but instruction-tuning on short texts (Wu et al., 2022; Gui et al., 2024b; Tang et al., 2024b; Lin et al., 2025) yields shallow supervision and struggles with long-document reasoning. Reinforcement learning (RL) offers a stronger alternative by refining models with rewards (Shao et al., 2024; Xie et al., 2025), enhancing outcome correctness and step-wise reasoning. Building on this (Trung et al., 2024; Liu et al., 2025), we propose RL-enhanced lightweight models with a two-phase, dual-reward scheme to bridge the accuracy–efficiency gap.

## 6 CONCLUSION

In this work, we present LITECOST, a reinforcement learning-enhanced framework that fine-tunes lightweight small language models (SLMs) to generate high-quality structured output for long-document QA. Through Chain-of-Structured-Thought (CoST) procedure and Group Relative Policy Optimization (GRPO), LITECOST enables a 3B-scale model to approach GPT-4o-mini and 7B models to achieve GPT-4o–level performance, while substantially reducing inference latency and resource consumption. We further discuss the potential applications of LITECOST across diverse domains in Appendix E. This works demonstrate the potential of scalable, cost-efficient long-document QA and paves the way for effective LLM reasoning grounded in structured representations.

**Limitations.** While LITECOST demonstrates strong performance across financial, legal, and open-domain QA, its generalization to other distinct domains remains to be fully verified. Despite the current scarcity of domain-specific document QA datasets suitable for constructing training data, we reserve the exploration of broader domain adaptation and more diverse QA scenarios for future work.

ACKNOWLEDGEMENT

This work is supported by National Key R&D Program of China under Grant No.2024YFA1012700, Guangdong provincial project 2023CX10X008, NSF of China (62402409), Youth S&T Talent Support Programme of Guangdong Provincial Association for Science and Technology (SKXRC2025461), and the Young Talent Support Project of Guangzhou Association for Science and Technology (QT-2025-001).

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

# Appendix Contents

> **The Prompt for Structure Selection**
>
> This is a data structure selection task. Based on the given `question`, choose the most
> suitable data structure to answer the question. You can choose from the following options:
> - Table (statistical comparisons, multi-source data)
> - Graph (entity connections, network analysis)
> - Text chunks (simple facts, single-step queries)
> Return your answer in format: {"answer": "data structure", "reason": "concise explanation"}
>
> The question is: {question}
>
> Guidelines:
> 1. If the question requires aggregating and comparing numbers/attributes from multiple
> sources -> Use Table
> 2. If the question focuses on connections between entities -> Use Graph
> 4. If the question can be answered with direct text extraction -> Use Text chunks

Figure 8: The prompt for selecting the most optimal structure.

## A  IMPLEMENTATION OF CHAIN-OF-STRUCTURED-THOUGHT (CoST)

During the entire *CoST: Structure-First Reasoning and Trace Generation* process, we construct
prompts in four key processes respectively. Firstly, in the structure analysis stage, we construct the
prompt about structure selection and schema construction, as shown in Fig. 8 and Fig. 9. Secondly,
we adopt an instruction-based chain-of-thought paradigm that performs step-by-step reasoning to
progressively extract structured knowledge and generate the CoST trace, as shown in Fig. 10. Finally,
in order to obtain high-quality *serialized structured output (SSO)*, we conduct the quality verification,
followed by iterative refinement. The details are shown in Fig. 11, 12.

### A.1  THE PROMPT OF STRUCTURE ANALYSIS

To dynamically select appropriate data structures and perform question preprocessing, we conduct
structure analysis, which includes both structure selection and schema construction. On one hand, we
design prompts that enable question-oriented structure selection. On the other hand, we construct an
accurate, task-specific schema through careful preprocessing of the question, before instruction-based
information extraction. The prompts of structure analysis are shown in Fig. 8, 9.

### A.2  THE PROMPT OF CoST TRACE GENERATION

To generate high-quality trace and structured output, we adopt an schema guided chain-of- thought
paradigm composed of: (1) step-by-step task instructions, (2) input text, and (3) a schema dynamically
generated from the question. GPT-4o is prompted with these schema-informed instructions to produce
intermediate reasoning traces, which are then used to supervise instruction-tuned models in zero-shot
or few-shot settings. Fig. 10 illustrates this reasoning process for the structured output (*e.g.,* table) .

### A.3  THE PROMPT OF QUALITY VERIFICATION

To evaluate the quality of the generated structured output, we employ an LLM-as-Judge framework.
Given the original question and the model-generated answer derived from the extracted structure,
we prompt GPT-4o to assess whether the answer correctly addresses the question. An inference
is considered correct and retained only if it exactly matches the expected answer based on the
prompt instructions. The prompt includes the original question, the extracted result, and evaluation
instructions guiding the model to make a binary decision, as detailed in Fig. 11.

**The Prompt for Schema Construction**

You are a schema generation assistant. Given a query, analyze its task and generate a structured schema to guide entity extraction from documents. The schema should include all relevant entities and their types, even for simple queries.

### Key Rules:
1. **Entity Extraction**:
   - If the query involves a single entity (e.g., "Find the company with the highest revenue"), extract the entity type (e.g., "Company", "Revenue").
   - If the query contains composite entities (e.g., mathematical formulas or nested metrics), split them into separate entities (e.g., "Accounts Payable", "COGS", "Inventory").
2. **Language Support**:
   - The schema should support both Chinese and English queries. Use the language of the query for entity names.
3. **Output Format**:
   - Ignore any output requirements in the query. The output must strictly follow this format: (EntityType1, EntityType2, ...).
   - Adhere strictly to the format: (EntityType1, EntityType2, ...).
   - The schema should include only entity types, listed in order and separated by commas.
   - Entity types should be clear and specific (e.g., "Company", "Revenue", "Year").

### Examples:
{examples}
### Real Data:
Query: {query}

### Output:\

Figure 9: The prompt for dynamically constructing schema.

**The Prompt for Table Extraction**

# Task Objective
Construct a structured table from raw text content that strictly adheres to the provided schema.

# Input Schema
Schema Columns: {schema}

# Step-by-Step Instructions
Step 1: **Entity Extraction**:
   - Identify relevant entities from raw content based on the schema
   - Ensure extracted entities align with schema column definitions
   - Generate an intermediate result listing all extracted entities.
Step 2: **Entity Linking**:
   For each row in raw content:
   - Map values to schema columns using exact keyword matching
   - Directly link the raw entities as they appear, without performing any additional inference.
   - If no direct match, Use contextually related values marked with [INFERRED].
   - Generate an intermediate result that outputs the relationships between entities (in the form of tuples), explaining how each entity is connected.
Step 3: **Summarization**:
   - Based on the entities and linking relationships from the previous steps, summarize the information and organize it into a table
structured format that conforms to the Schema.
   - Generate a draft table as an intermediate result, ensuring that each Schema column is correctly populated.
Step 4: **Final Output Formatting**:
   Follow this exact structure:
   ("table"{tuple_delimiter}<Title>{tuple_delimiter}<Source>{tuple_delimiter}<Description>){record_delimiter}
   ("header"{tuple_delimiter}<COLUMN_1>{tuple_delimiter}<COLUMN_2>...){record_delimiter}
   ("row"{tuple_delimiter}<VALUE_1>{tuple_delimiter}<VALUE_2>...){record_delimiter}
   ...
   {completion_delimiter}

# Critical Rules
- STRICTLY FOLLOW SCHEMA COLUMN ORDER
- NEVER invent data not present in raw content
- **DO NOT perform any additional computation or reasoning; simply perform extraction.**
- **The chain-of-thought reasoning for each step should be clearly documented as intermediate results before generating the final table.**

# Current Task
Schema: {schema}
Raw Content: {content}

# Output Structure:\

Figure 10: The prompt for extracting table step-by-step.

---

**The Prompt for Data Verification**

Given a standard answer, and a provided answer, your task is to verify if the provided answer is correct according to the standard answer.
You should base your evaluation strictly on the content and correctness of the provided answer in relation to the standard answer.

Here are the details:
<standard answer>
 {std_ans}
<standard answer>

<provided answer>
 {ans}
<provided answer>

## Important Notes:
- 1. If the <provided answer> is "I don't know", "I can't get the answer", or any similar phrase, consider the answer as incorrect.
- 2. If the <provided answer> is vague, irrelevant, or does not answer the question properly, return "False".
- 3. If the <standard answer> is a numerical value, and the format of the <provided answer> is different from that of the <standard answer> , but the numerical values are the same, then it is considered that the meanings are consistent. For example, if the <standard answer> is 0.98 and the <provided answer> is 98%, it is considered that the meanings are consistent, return "True".
- 4. If the <standard answer> is a numerical value, and the final result of the <provided answer> is consistent with the <standard answer> after rounding, then it is considered that the meanings are consistent. For example, if the <standard answer> is 2 and the <provided answer> is 1.98, it is considered that the meanings are consistent, return "True".

## Output Format:
Please respond with the result in the following format:
{{check: True or False}}
Do not provide any additional explanation or context, just return the result in the specified format.

---

Figure 11: The prompt for verifying the data quality.

### A.4 THE PROMPT OF ITERATIVE REFINEMENT

To enhance Group Relative Policy Optimization (GRPO) training with more challenging learning signals, we introduce an Iterative Structuralizer module that refines low-quality samples through recursive structured knowledge regeneration. The prompt for refining structured data extraction (*e.g.*, table) is detailed in Fig. 12. Formally, this process is implemented as a recursive function over the evolving extraction state, gradually improving coverage and accuracy across iterations:

$$S^{(t+1)} = \begin{cases} S^{(t)}, & \text{if } \mathcal{K}(S^{(t)}, q) = \texttt{True} \\ f_{\text{extract}}(q, c, S^{(t)}), & \text{otherwise,} \end{cases} \tag{5}$$

where $\mathcal{K}(S^{(t)}, q)$ is a sufficiency evaluator that returns `True` if the current structured knowledge can answer the question; $f_{\text{extract}}$ is the structured knowledge extraction function, and $c$ is the context. The process terminates when $\mathcal{S}(K^{(t)}, q) = \texttt{True}$ or when a predefined maximum number of iterations is reached. The final structured knowledge $S^*$ is then used for downstream reasoning.

## B ADDITIONAL DETAILS OF REINFORCE LEARNING

### B.1 GROUP-RELATIVE ADVANTAGE AND KL DIVERGENCE

For completeness, we provide additional details of the GRPO optimization. The importance sampling ratio is defined as $r_i^{\text{ratio}} = \frac{\pi_\theta(o_i|\mathbf{v})}{\pi_{\theta_{\text{old}}}(o_i|\mathbf{v})}$, which quantifies the relative likelihood of generating output $o_i$ under the new policy compared with the old policy $\pi_\theta$. The group-relative advantage $A_i$ is calculated based on the relative rewards of outputs within the same group only.

To ensure stable optimization, the clipping operator $\text{clip}(r_i^{\text{ratio}}, 1 - \epsilon, 1 + \epsilon)$ restricts the update magnitude within the trust region $[1 - \epsilon, 1 + \epsilon]$, thereby avoiding destabilizing large parameter changes. Finally, taking the minimum between the unclipped term $r_i^{\text{ratio}} A_i$ and its clipped counterpart enforces a conservative update, which balances aggressive improvements with training stability.

---

**The Prompt for Data Refinement**

Reconstruct the table by combining Original Data and Current Table while STRICTLY MAINTAINING the existing format structure.

\# Format Requirements
1. Follow this EXACT structure:
   ("table"{tuple_delimiter}<Title>{tuple_delimiter}<Source>{tuple_delimiter}<Description>)
{record_delimiter}
   ("header"{tuple_delimiter}<COLUMN_1>{tuple_delimiter}<COLUMN_2>...){record_delimiter}
   ("row"{tuple_delimiter}<VALUE_1>{tuple_delimiter}<VALUE_2>...){record_delimiter}
   ...
   {completion_delimiter}
2. Strict Rules:
   - PRESERVE original column order and headers
   - USE EXISTING title/source/description unless new metadata found
   - DO NOT add/remove columns
   - ONLY modify rows with missing/conflicting data

\# Input Context
Original Data:
{content}

Current Table Structure:
{current_table}

\# Output:\

Figure 12: The prompt for refining low-quality data.

In addition, the KL divergence term $D_{\mathrm{KL}}(\pi_\theta \| \pi_{\mathrm{ref}})$ plays a critical role in regularizing policy updates. It penalizes large deviations from a stable reference policy $\pi_{\mathrm{ref}}$, preventing overfitting to noisy reward signals and maintaining alignment with the base model's distribution. The coefficient $\beta$ controls the strength of this regularization, striking a balance between exploration (deviation from the reference) and stability (consistency with the pretrained policy). This regularization is particularly important when rewards are sparse or noisy, such as process rewards, as it prevents the model from overfitting to unstable signals while still enabling gradual improvement (Shao et al., 2024).

### B.2 THE PROMPT OF ANSWER COMPLETENESS

To measure the semantic similarity between the generated structured outputs and the ground truth during the GRPO process, we employ GPT-4o-mini as an automatic evaluator. The evaluator examines the outputs along four dimensions: null check, core field coverage, semantic alignment, and semantic equivalence—and then produces a score within the range [0, 100]. This score, denoted as $\mathcal{S}_{\mathrm{sem}}$ in Equation 5, provides an accurate quantification of the semantic similarity of the structured data, with the prompting details shown in Fig. 13.

## C EXPERIMENTAL DETAILS

### C.1 EVALUATION METRICS

Following the approach of Loong (Wang et al., 2024), we prompt GPT-4o as a judge to against the golden answer and question requirements in three dimensions: Accuracy, Hallucinations, and Completeness, on a 0–100 scale, as detailed in Figure 15. With this evaluation method, the Judger model would output a percentage score along with its corresponding explanation. Given the limited ground-truth annotations for information extraction on long-context documents, we adopt a 2-hop evaluation following the principle of STRUCTSUM (Jain et al., 2024), which uses the structured outputs to answer QA pairs derived from the input text. Our core intuition is that if an LLM can accurately answer questions based solely on the extracted data, the extraction process has likely preserved the essential information-thus reflecting its quality.

```
The Prompt for Semantic Verification

[Task Description]
Act as a semantic consistency evaluator. Assess whether [Generated Data] aligns with [Ground Truth Data] in
content and meaning. Evaluate step by step, then output a score (0–100), key_differences, and warnings.

[Steps]
1. Null Check
- If 【Generated Data】 is empty or trivial (e.g., "N/A"), return score = 0 with warning "Generated data is empty".
2. Core Field Coverage
- Check if key fields (e.g., company, revenue, time) in ground truth are covered.
- Missing core fields greatly lowers the score; non-critical fields have less impact.
- Different names with the same meaning (e.g., "Revenue" vs "Operating Income") count as equivalent.
3. Semantic Alignment
- Verify if fields express the same meaning in:
  - Numbers (10M vs 10,000,000)
  - Units/ratios/time spans
  - Entities (Microsoft vs Microsoft Corp.)
  - Time expressions
  - Factual relations (Founder is Jack Ma vs Jack Ma founded the company)
4. Semantic Equivalence (No Penalty)
- Acceptable variations: number formatting, abbreviations, time formats, synonyms.

[Ground Truth Data]
{Ground Truth}
[Generated Data]
{Generated Data}

[Output Format]
Return JSON only:
{{
"score": 0-100,
"key_differences": ["..."],
"warnings": ["..."]
}}
```

Figure 13: The prompt for verifying the similarity of generated output and ground truth.

For experiments on LongBench (Bai et al., 2024), we additionally use the standard F1 score to assess the quality of reasoning based on the extracted information. Consistent with the Loong setup, GPT-4o is employed as the reasoning agent to generate final answers from the structured outputs.

## C.2 DETAILS OF EXPERIMENTAL SETTING.

**Base Models.** To comprehensively evaluate the extraction ability of LITECoST, we compare it with several state-of-the-art models, including Llama3.2-3B-Instruct (Grattafiori et al., 2024), Qwen2-7B-Instruct (Yang et al., 2024), Llama-3.1-8B-Instruct, Qwen2.5-14B-Instruct, GPT4o-mini, GPT-4o (Achiam et al., 2023), and Deepseek-R1 (Guo et al., 2025). For fairness, all models are evaluated under the same prompting setup, where each is guided by a task-specific schema derived from the given question or instruction. To avoid exceeding the context window, all models perform document-level extraction and merged the resulting structured sub-knowledge.

**Fine-tuned IE Models.** For comparison with fine-tuned IE models, both LITECoST and the baselines are trained on our CoST-curated dataset under identical conditions. We then evaluate their extraction performance to highlight differences in fine-tuning strategies and demonstrate the relative advantages of our approach.

**Modular Frameworks.** We further consider modular extraction frameworks such as StructRAG (Li et al., 2025c). In this setup, we configure the Router and Structurizer components with the same backbones (*i.e.,* LLaMA-3.2-3B-Instruct and Qwen2-7B-Instruct), while employing GPT-4o as the Utilizer for reasoning. This setup ensures that the comparison between StructRAG and our LITECoST remains both fair and rigorous.

**Baseline Prompting Template.** To ensure fairness, all baselines and our model use the same input prompting format for structured extraction, as shown in Fig. 14. The LLM baselines are evaluated in a zero-shot setting, and entries labeled "LLM baseline" refer to this zero-shot performance. Our LITECoST-tuned models are compared directly against these baselines because both rely on the identical prompt template, ensuring that performance differences arise from CoST distillation and the two-stage training procedure rather than from prompting variations or model-capacity differences.

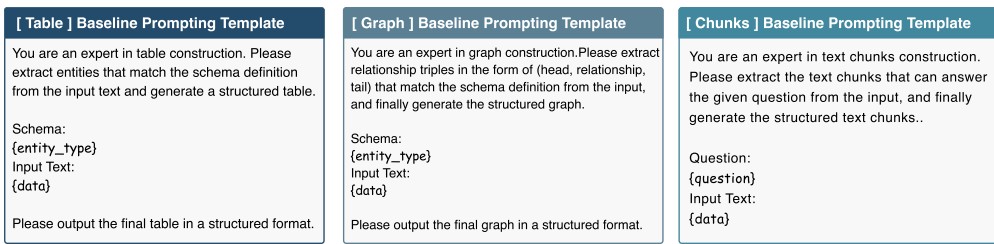

Figure 14: The baseline prompt template for different structured extraction.

Table 6: The performance of various LLMs under different prompting settings, showing *Average Scores (AS, 0-100)* and *Perfecr Rate (PR, 0-1)* on the *Finance* subset of Loong benchmark.

| Backbone | Method | Spotlight Locating | | Comparison | | Clustering | | Chain of Reasoning | | Overall | |
|---|---|---|---|---|---|---|---|---|---|---|---|
| | | AS | PR | AS | PR | AS | PR | AS | PR | AS | PR |
| Qwen2.5-14B-Ins | Zero-Shot | 83.74 | 0.57 | **82.12** | 0.56 | 69.96 | 0.24 | 66.41 | 0.10 | 75.60 | 0.38 |
| | CoT (Wei et al., 2022) | 85.93 | 0.63 | 81.38 | 0.57 | 73.28 | 0.30 | 67.70 | **0.26** | 77.51 | 0.44 |
| | Dspy (Khattab et al., 2023) | 80.90 | 0.55 | 80.90 | **0.60** | **76.86** | 0.34 | 63.60 | 0.21 | 76.99 | 0.44 |
| | LITECOST (CoST) | **88.03** | **0.72** | 80.45 | 0.56 | 76.72 | **0.37** | **68.35** | 0.25 | **79.01** | **0.48** |
| GPT-4o | Zero-Shot | 84.10 | 0.73 | 80.53 | 0.60 | 81.50 | 0.50 | 64.30 | 0.25 | 79.32 | 0.54 |
| | CoT (Wei et al., 2022) | 85.57 | 0.69 | 84.40 | **0.66** | 79.58 | 0.44 | 64.45 | **0.30** | 80.08 | 0.54 |
| | Dspy (Khattab et al., 2023) | 86.17 | 0.74 | 82.65 | 0.62 | 80.00 | 0.47 | 67.25 | 0.27 | 80.26 | 0.54 |
| | LITECOST (CoST) | **88.80** | **0.75** | **84.50** | 0.65 | **82.24** | **0.50** | **68.90** | 0.28 | **82.39** | **0.56** |

## C.3 NUMERICAL RESULTS OF THE RADAR CHART

Table 6 provides the detailed numerical results on the Finance subset of Loong across backbone LLMs. Relative to Zero-Shot prompting, Chain-of-Thought (CoT) consistently improves overall performance, raising the average score (AS) from 75.60 to 77.51 on Qwen2.5-14B-Ins and from 79.32 to 80.08 on GPT-4o. Building further on CoT, our CoST-based LITECOST achieves the strongest results across all backbones, with overall AS/PR reaching 79.01/0.48 on Qwen2.5-14B-Ins, and 82.39/0.56 on GPT-4o. The improvements are consistent across subtasks—for example, CoST yields a gain of +6.62 AS on Clustering with GPT-4o and +8.62 AS on Spotlight Locating with Qwen2.5-14B-Ins, demonstrating its robustness in enhancing reasoning-intensive extraction.

We have also integrate prompt-optimization methods, including DSPy (Khattab et al., 2023), into our evaluation. Specifically, we use the same zero-shot structured-extraction instructions as the initial input prompts for GPT-4o and Qwen2.5-14B-Instruct, and then apply the MiPRO optimizer to refine these prompts—automatically generating Chain-of-Thought–style reasoning steps and optimized structured prompts. MiPRO is conditioned on examples drawn from a randomly sampled 10% subset of the LITECOST training data (finance). We observe that although DSPy provides slight improvements over the zero-shot and CoT baselines, our method still performs substantially better. This highlights both the effectiveness and robustness of CoST, and further illustrates the challenges of applying automated prompt optimization to structure-aware extraction tasks.

## C.4 ROBUST LONG-DOCUMENT HANDLING.

The results in Fig. 16 show that both Chain-of-Structured-Thought (CoST) and LITECOST exhibit strong robustness as document length increases.

**LLM+CoST.** The overall score of LLM equipped with CoST decreases by only 25.21 points when moving from Set1 (91.46) to the most challenging Set4 (66.25), whereas zero-shot prompting exhibits a much sharper decline (92.33 → 58.42) and CoT similarly drops from 87.90 to 60.49. On Set4, CoST surpasses zero-shot and CoT by 5.76 and 7.83 points, respectively, indicating that CoST provides substantially greater robustness under increasing difficulty.

**LITECOST-tuned SLM.** Our LITECOST-tuned SLM also demonstrates strong robustness, exhibiting only a modest 24.46-point decline as document length increases. Even on the most challenging Set4, it achieves a score of 64.89, outperforming all LLM-based baselines: +6.47 over GPT-4o, +4.77 over Qwen2.5-14B-Instruct, and +12.01 over LLaMA-3.1-8B-Instruct.

**The Prompt for LLM Score Evaluation**

[Gold Answer]
{std_ans}

[The Start of Assistant's Predicted Answer]
{ans}
[The End of Assistant's Predicted Answer]

[System] We would like to request your feedback on the performance of the AI assistant in response to the user question displayed above according to the gold answer. Please use the following listed aspects and their descriptions as evaluation criteria: - Accuracy and Hallucinations: The assistant's answer is semantically consistent with the gold answer; The numerical value and order need to be accurate, and there should be no hallucinations. - Completeness: Referring to the reference answers, the assistant's answer should contain all the key points needed to answer the user's question; further elaboration on these key points can be omitted. Please rate whether this answer is suitable for the question. Please note that the gold answer can be considered as a correct answer to the question. The assistant receives an overall score on a scale of 1 to 100, where a higher score indicates better overall performance.Please note that if the assistant's answer and the gold answer fully meet the above criteria, its overall rating should be the full marks (100). Please first provide a comprehensive explanation of your evaluation, avoiding any potential bias.Then, output a line indicating the score of the Assistant. PLEASE OUTPUT WITH THE FOLLOWING FORMAT, WHERE THE SCORE IS A SCALE OF 1 TO 100 BY STRICTLY FOLLOWING THIS FORMAT: "[[score]]", FOR EXAMPLE "Rating: [[100]]":

<Start Output>
Evaluation evidence: your evluation explanation here, no more than 100 words Rating: [[score]]
<End Output>
Now, start your evaluation:

Figure 15: The prompt for evaluation using LLM Score.

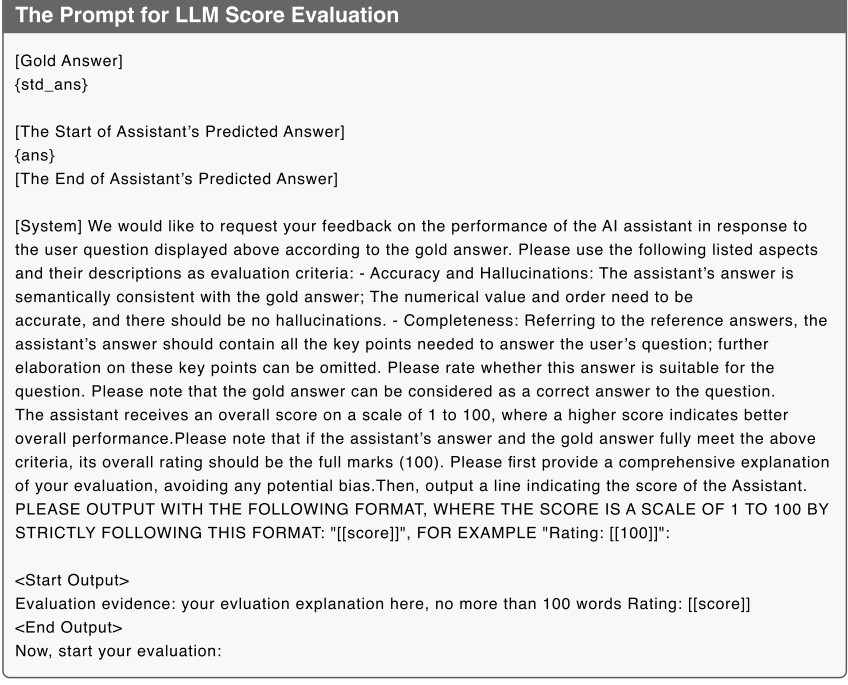

Figure 16: Performance (LLM score) versus document-set size for (a) CoST with baseline methods (zero-shot and CoT) and (b) LiteCoST-tuned models with different base models.

These results underscore the particular strength of CoST and LITECOST in handling long-document inputs, consistently maintaining high reasoning accuracy even as context length increases.

### C.5    FULL RESULTS OF REASONING W/ STRUCTURED DATA.

By selecting the optimal structure for each QA sample in the Loong benchmark, the dataset comprises 823 tables, 400 graphs, and 377 text chunks. The purpose of this experiment is to explore how *serialized structured output (SSO)* contributes to model performance on knowledge-intensive reasoning tasks. Therefore, we curate such structures through our LITECOST framework with GPT-4o as the base model, and then evaluate the results of LLMs reasoning over these structured representation.

Table 8 shows that all models achieve better reasoning performance when leveraging the extracted structured information, compared to the baseline that directly processes the raw long-form documents, underscoring the necessity of accurate and effective structured knowledge extraction.

**Pros.** The gains are particularly notable in comparison and clustering tasks, where all models show improvements in LLM scores, along with at least 0.11 and 0.17 increases in Perfect Rate, respectively. These results highlight the value of structured knowledge in capturing entity relationships and aggregating discrete information for complex reasoning.

**Cons.** We observe slight performance drops in certain subtasks (*Spotlight Locating* and *Chain-of-Reasoning*) after integrating structured data. These results reflect a trade-off between structured clarity and the flexibility of unstructured reasoning:

- **Advantages of structured data:** Improves aggregation and clustering tasks by enabling more effective comparison and summarization.
- **Challenge in Spotlight Locating:** Fine-grained localization may degrade when structured data omits subtle contextual cues, reducing recall.
- **Challenge in Chain-of-Reasoning:** Complex multi-step inference may lose nuance when context is compressed into structured formats.

Here, we provide a case to illustrate this issue with the following Spotlight Locating task:

- **Question:** What is the name of the company with $30, 179$ in Accounts Receivable?
- **Ground Truth:** CIRTRAN CORP
- **LLM Answer:** The information provided in the context does not list a company with $30, 179$ in Accounts Receivable. Therefore, the answer is $\boxed{\text{Not Provided}}$

Table 7: Extracted table for the given query.

| Company | Accounts Receivable |
|---|---|
| AXIM Biotechnologies, Inc. | $23,642 |
| CIRTRAN CORP. | None |
| Arax Holdings Corp. | $453,837 |
| High Wire Networks, Inc. | $4,483,1 |

In this case, due to value corruption or omission during extraction, CIRTRAN CORP's relevant numerical value is missing in the structured table, leading the model to respond incorrectly. A large language model performing direct reasoning over the full text, in contrast, can flexibly match partial evidence (e.g., approximate figures or nearby mentions) without being constrained by structure.

### C.6 Computational Resource.

**Model Deployment.** We deployed and run models ranging from 3B to 14B model size on a cluster equipped with eight NVIDIA RTX 4090 GPUs, each with 24GB of VRAM. For closed sourced large language models such as GPT and Claude, we accessed them via API calls.

**Computational cost.** Our cost mainly comes from two components:

- **Data Generation.** This includes structure analysis, CoST trace generation, quality verification, and iterative refinement using GPT-4o. The average cost is approximately **$30** per domain.
- **GRPO Fine-tuning.** Reward computation relies on GPT-4o-mini to evaluate structural alignment, format compliance, and answer correctness. This adds an additional **$10** per training run.

The total cost of **$40** is necessary and acceptable because it amortizes extremely well: once LiteCoST is trained, downstream inference relies solely on compact SLMs, eliminating repeated LLM calls and yielding substantial savings during deployment.

## D   Case Study on Practical Long-Document QA Tasks.

### D.1   Case Study on RL-Enhancement

As shown in Fig. 17, we present a representative long-context QA example to evaluate the model's information extraction capabilities. This question is particularly challenging, as it requires the model to retrieve multiple pieces of evidence from the input text and accurately integrate them into coherent structured information. Specifically, the case study includes the question and documents input (grey box), predictions from the base LLaMA-3.2-3B-Instruct model (blue box), its finetuned variant (orange box) and our LITECOST model (green box).

In addition, incorrect predictions are highlighted in red, while correct ones are marked in green. The results show that the Llama base model performs poorly at information extraction and fails

Table 8: Structured data quality evaluation of LITECOST via Chain-of-Structured-Thought (CoST) on the Loong benchmark, compared against popular LLMs. *SD* denotes structured data.

| Model | Context Length | *Spotlight Locating* | | *Comparison* | | *Clustering* | | *Chain of Reasoning* | | *Overall* | |
|---|---|---|---|---|---|---|---|---|---|---|---|
| | | *AS* | *PR* | *AS* | *PR* | *AS* | *PR* | *AS* | *PR* | *AS* | *PR* |
| Qwen2-72B-Ins | 128k | 54.17 | 0.36 | 42.38 | 0.20 | 36.71 | 0.04 | 47.76 | 0.18 | 43.29 | 0.15 |
| SD → Qwen2-72B-Ins | 128k | 57.30 | 0.31 | 54.80 | 0.38 | 61.46 | 0.23 | 46.36 | 0.16 | **55.70** | **0.25** |
| GPT-4o-mini | 128k | 59.46 | 0.49 | 51.90 | 0.27 | 34.55 | 0.04 | 64.28 | 0.39 | 49.25 | 0.24 |
| SD → GPT-4o-mini | 128k | 63.23 | 0.44 | 53.04 | 0.38 | 59.63 | 0.21 | 55.98 | 0.26 | **58.02** | **0.29** |
| GPT-4o | 128k | 73.95 | 0.62 | 50.50 | 0.28 | 44.29 | 0.09 | 57.05 | 0.28 | 53.47 | 0.26 |
| SD → GPT-4o | 128k | 62.11 | 0.33 | 63.27 | 0.41 | 68.06 | 0.29 | 53.52 | 0.22 | **62.51** | **0.30** |
| Claude3.5-Sonnet | 200k | 58.45 | 0.49 | 54.21 | 0.25 | 45.77 | 0.07 | 43.92 | 0.25 | 48.85 | 0.23 |
| SD → Claude3.5-Sonnet | 200k | 47.60 | 0.34 | 54.64 | 0.41 | 66.95 | 0.31 | 50.45 | 0.23 | **57.32** | **0.31** |

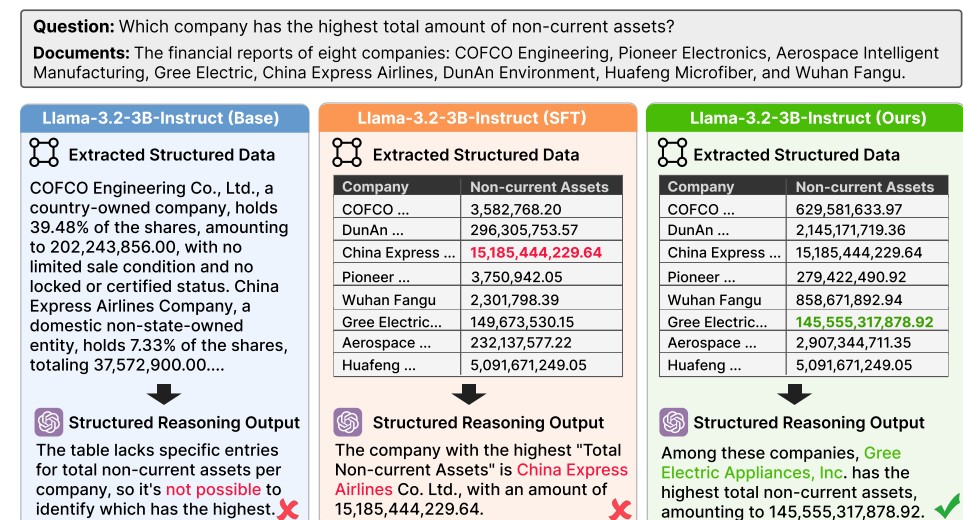

Figure 17: Case study on representative information extraction tasks for large language models, comparing the information extraction capabilities of three Llama-3.2-3B-Instruct variants.

to organize content into structured formats such as tables. The fine-tuned model exhibits partial extraction ability but remains inaccurate in long-context, multi-document settings. In contrast, our LITECOST accurately integrates dispersed information into high-quality structured tables, with gains largely attributed to supervised CoT reasoning and reinforcement learning. As shown in Fig. 17, LITECOST provides significantly enhanced interpretability compared to its base model.

## E  GENERALIZATION ON DIFFERENT DOMAINS

### E.1  LEGAL DOMAIN.

To examine the generalization capabilities of LITECOST, we apply it to the legal domain by curated by from LegalBenchRAG (Pipitone & Alami, 2024), a dataset of 6,858 query-answer pairs over a corpus of over 79M characters, entirely human-annotated by legal experts. Leveraging the training data described in Sec. 4.1, we perform RL fine-tuning on small language models. The fine-tuned models are subsequently evaluated on the legal subset of Loong (Wang et al., 2024) to assess their ability to generate structured outputs in complex legal contexts.

Table 4 shows that our LITECOST-tuned 3B/7B models, LLaMA-LiteCoST and Qwen-LiteCoST, achieve substantial improvements over their respective base models and deliver performance comparable to proprietary LLMs. From these results, we further derive two key insights:

(1) Notably, our method elevates LLaMA variant to surpass Qwen in overall performance. The two backbones, however, exhibit complementary strengths: LLaMA excels on the Spotlight Locating and Comparison subtasks, while Qwen achieves superior results on Clustering and Chain-of-Reasoning.

Table 9: An overview of the dataset statistics for a subset of LongBench. 'Source' denotes the origin of the context. 'Avg len' (average length) is computed using the number of words for the English (code) datasets and the number of characters for the Chinese datasets. 'Accuracy (CLS)' refers to classification accuracy, while 'Accuracy (EM)' refers to exact match accuracy.

| Dataset | ID | Source | Avg len | Metric | Language | #data |
|---|---|---|---|---|---|---|
| *Single-Document QA* | | | | | | |
| NarrativeQA | 1-1 | Literature, Film | 18,409 | F1 | English | 200 |
| Qasper | 1-2 | Science | 3,619 | F1 | English | 200 |
| *Multi-Document QA* | | | | | | |
| HotpotQA | 2-1 | Wikipedia | 9,151 | F1 | English | 200 |
| 2WikiMultihopQA | 2-2 | Wikipedia | 4,887 | F1 | English | 200 |

(2) Qwen2-7B-Instruct performs exceptionally well on the legal domain, even surpassing its 14B counterpart, potentially due to: ❶ stronger domain adaptation from a larger proportion of legal corpora in training; ❷ the tendency of larger models to over-generate in long, highly structured legal texts, leading to hallucinations or format drift, whereas the smaller 7B model more faithfully adheres to schemas and maintains consistency.

### E.2 OPEN-DOMAIN QA.

Beyond the Loong benchmark, we further evaluate our method on LongBench (Bai et al., 2024), a widely recognized multi-task benchmark for long-document QA that covers key real-world application scenarios across literature, science, encyclopedias, etc. These datasets contain far more nuanced information. As shown in Table 5, our LITECOST effectively extends to settings with richer and more subtle semantics, demonstrating strong generalization beyond strictly structured environments. Our analysis focuses on both single- and multi-doc QA tasks across four datasets, as shown in Table 9.

**Single-Doc QA.** For single-document QA, we focus on datasets containing longer and more challenging documents. We evaluate on NarrativeQA (Kočiskỳ et al., 2018), consisting of full-length stories paired with questions designed to test deep reading comprehension. We also include Qasper (Dasigi et al., 2021), a dataset featuring QA over NLP research papers, annotated by domain experts.

**Multi-Doc QA.** Multi-document QA requires models to extract and combine information from several documents to obtain the answer, which is usually more challenging than single-doc QA. We evaluate on two Wikipedia-based multi-hop QA datasets: HotpotQA (Yang et al., 2018), 2WikiMultihopQA (Ho et al., 2020). HotpotQA involves a number of 2-hop questions directly written by native speakers given two related paragraphs. 2Wiki-MultihopQA consists of up to 5-hop questions that are synthesized through manually designed tem- plates to ensure that they cannot be solved through shortcuts. Each question in the original datasets is supplemented by 2-4 supporting paragraphs that provide one-step reasoning evidence and several distracting paragraphs.

### E.3 OTHERS.

Beyond the finance, legal, and scientific QA tasks, LITECOST can be applied to a wide range of fields, enabling effective and efficient structured extraction from large-scale unstructured corpora:

- **Healthcare**: extracting patient attributes, treatment outcomes, and adverse event reports to enhance clinical decision-making and pharmacovigilance.
- **Scientific literature**: supporting literature understanding, analysis, and question answering through the extraction of experimental settings, results, and methodological details.
- **Policy and government**: structuring entities and relations from legislative and regulatory documents to facilitate compliance monitoring and policy evaluation.
- **Enterprise analytics**: organizing information from reports, manuals, and support logs into structured forms to improve knowledge management and retrieval-augmented applications.

Together, these applications highlight the broad adaptability of LITECOST in transforming unstructured knowledge into structured representations that directly support diverse downstream tasks.

Table 10: Full results of Table 1. *AS* represents *Avg Scores (0~100)* and *PR* denotes *Perfect Rate* (0~1). **Bold** indicates the best result within each setting, and underlined indicates the second best.

| Model | Model Size | Spotlight Locating | | Comparison | | Clustering | | Chain of Reasoning | | Overall | |
|---|---|---|---|---|---|---|---|---|---|---|---|
| | | *AS* | *PR* | *AS* | *PR* | *AS* | *PR* | *AS* | *PR* | *AS* | *PR* |
| All Set (**10K-250K**) | | | | | | | | | | | |
| LLaMA-3.2-3B-Instruct | 3B | 49.90 | 0.16 | 52.10 | 0.14 | 47.89 | 0.07 | 46.85 | 0.06 | 49.37 | 0.11 |
| Qwen2-7B-Instruct | 7B | 63.10 | 0.36 | 67.85 | 0.37 | 60.83 | 0.18 | 52.25 | 0.09 | 62.10 | 0.26 |
| LLaMA-3.1-8B-Instruct | 8B | 55.03 | 0.20 | 51.60 | 0.15 | 51.50 | 0.04 | 44.75 | 0.02 | 51.32 | 0.10 |
| GPT-4o-mini | 8B | 84.42 | 0.70 | 80.40 | 0.67 | 77.38 | 0.40 | 65.35 | 0.18 | 78.08 | 0.51 |
| Qwen2.5-14B-Instruct | 14B | 83.74 | 0.57 | 82.12 | 0.56 | 69.96 | 0.24 | 66.41 | 0.10 | 75.60 | 0.38 |
| GPT-4o | 200B | 84.10 | 0.73 | 80.53 | 0.60 | 81.50 | 0.50 | 64.30 | 0.25 | 79.32 | **0.54** |
| Deepseek-R1 | 671B | 84.27 | 0.62 | 78.97 | 0.55 | 75.42 | 0.34 | 74.40 | 0.35 | 78.18 | 0.46 |
| LLaMA-3.2-3B-Instruct (*SFT*) | 3B | 74.39 | 0.45 | 75.53 | 0.45 | 73.64 | 0.29 | 59.15 | 0.12 | 72.27 | 0.35 |
| LLaMA-3.2-3B-Instruct (*Ours*) | 3B | 81.27 | 0.53 | 78.08 | 0.49 | 78.34 | 0.36 | 64.75 | 0.16 | 76.95 | 0.40 |
| Qwen2-7B-Instruct (*SFT*) | 7B | 82.23 | 0.58 | 81.15 | 0.56 | 75.91 | 0.33 | 62.40 | 0.11 | 76.83 | 0.42 |
| Qwen2-7B-Instruct (*Ours*) | 7B | 83.97 | 0.62 | 81.55 | 0.59 | 81.00 | 0.43 | 67.98 | 0.18 | **79.93** | 0.48 |
| Set1 (**10K-50K**) | | | | | | | | | | | |
| LLaMA-3.2-3B-Instruct | 3B | 54.13 | 0.17 | 43.33 | 0.13 | 44.25 | 0.07 | 55.50 | 0.10 | 47.28 | 0.12 |
| Qwen2-7B-Instruct | 7B | 73.26 | 0.48 | 80.17 | 0.53 | 68.25 | 0.28 | 65.00 | 0.30 | 72.52 | 0.40 |
| LLaMA-3.1-8B-Instruct | 8B | 46.09 | 0.04 | 35.50 | 0.00 | 47.38 | 0.00 | 40.50 | 0.00 | 42.96 | 0.01 |
| GPT-4o-mini | 8B | 96.09 | 0.91 | 93.00 | 0.90 | 86.62 | 0.70 | 75.50 | 0.40 | 89.51 | 0.78 |
| Qwen2.5-14B-Instruct | 14B | 95.00 | 0.74 | 87.00 | 0.63 | 84.45 | 0.53 | 84.30 | 0.20 | 87.53 | 0.57 |
| GPT-4o | 200B | 96.09 | 0.91 | 90.00 | 0.80 | 91.25 | 0.78 | 95.00 | 0.90 | **92.33** | **0.83** |
| Deepseek-R1 | 671B | 91.96 | 0.78 | 90.50 | 0.80 | 88.75 | 0.68 | 99.50 | 0.90 | 91.02 | 0.76 |
| LLaMA-3.2-3B-Instruct (*SFT*) | 3B | 80.43 | 0.61 | 83.10 | 0.60 | 82.70 | 0.47 | 67.00 | 0.30 | 80.79 | 0.52 |
| LLaMA-3.2-3B-Instruct (*Ours*) | 3B | 91.52 | 0.78 | 81.33 | 0.57 | 88.95 | 0.60 | 75.00 | 0.50 | 85.95 | 0.62 |
| Qwen2-7B-Instruct (*SFT*) | 7B | 91.30 | 0.83 | 89.17 | 0.73 | 81.75 | 0.53 | 81.50 | 0.20 | 86.02 | 0.62 |
| Qwen2-7B-Instruct (*Ours*) | 7B | 92.17 | 0.83 | 89.67 | 0.77 | 88.75 | 0.72 | 84.80 | 0.60 | 89.40 | 0.75 |
| Set2 (**50K-100K**) | | | | | | | | | | | |
| LLaMA-3.2-3B-Instruct | 3B | 41.62 | 0.10 | 57.16 | 0.19 | 57.13 | 0.13 | 44.88 | 0.05 | 52.61 | 0.13 |
| Qwen2-7B-Instruct | 7B | 80.00 | 0.62 | 71.13 | 0.45 | 63.11 | 0.24 | 58.75 | 0.12 | 67.61 | 0.35 |
| LLaMA-3.1-8B-Instruct | 8B | 55.88 | 0.23 | 55.15 | 0.20 | 51.44 | 0.03 | 48.75 | 0.03 | 52.86 | 0.11 |
| GPT-4o-mini | 8B | 96.12 | 0.90 | 88.93 | 0.80 | 86.07 | 0.58 | 64.62 | 0.28 | **85.09** | 0.65 |
| Qwen2.5-14B-Instruct | 14B | 88.88 | 0.68 | 88.67 | 0.68 | 76.11 | 0.28 | 64.25 | 0.20 | 80.10 | 0.45 |
| GPT-4o | 200B | 90.38 | 0.80 | 96.27 | 0.72 | 89.67 | 0.71 | 66.88 | 0.33 | 85.02 | **0.67** |
| Deepseek-R1 | 671B | 85.75 | 0.68 | 83.04 | 0.61 | 81.17 | 0.50 | 78.62 | 0.45 | 82.07 | 0.56 |
| LLaMA-3.2-3B-Instruct (*SFT*) | 3B | 82.83 | 0.57 | 77.93 | 0.49 | 79.09 | 0.40 | 65.12 | 0.15 | 77.07 | 0.42 |
| LLaMA-3.2-3B-Instruct (*Ours*) | 3B | 88.12 | 0.60 | 81.33 | 0.56 | 83.98 | 0.48 | 65.25 | 0.20 | 80.79 | 0.40 |
| Qwen2-7B-Instruct (*SFT*) | 7B | 87.88 | 0.70 | 84.73 | 0.63 | 80.53 | 0.41 | 66.12 | 0.20 | 80.67 | 0.40 |
| Qwen2-7B-Instruct (*Ours*) | 7B | 90.00 | 0.72 | 85.53 | 0.68 | 85.28 | 0.56 | 68.50 | 0.23 | 83.39 | 0.50 |
| Set3 (**100K-200K**) | | | | | | | | | | | |
| LLaMA-3.2-3B-Instruct | 3B | 54.42 | 0.22 | 51.24 | 0.09 | 44.06 | 0.02 | 45.14 | 0.09 | 48.67 | 0.10 |
| Qwen2-7B-Instruct | 7B | 56.00 | 0.30 | 63.73 | 0.28 | 59.63 | 0.13 | 45.57 | 0.03 | 58.08 | 0.20 |
| LLaMA-3.1-8B-Instruct | 8B | 57.92 | 0.27 | 55.13 | 0.17 | 52.06 | 0.04 | 39.71 | 0.00 | 52.63 | 0.13 |
| GPT-4o-mini | 8B | 84.00 | 0.77 | 77.40 | 0.57 | 71.78 | 0.22 | 64.57 | 0.09 | 75.25 | 0.43 |
| Qwen2.5-14B-Instruct | 14B | 88.93 | 0.68 | 76.93 | 0.49 | 63.39 | 0.16 | 64.14 | 0.00 | 73.29 | 0.35 |
| GPT-4o | 200B | 88.33 | 0.83 | 76.13 | 0.49 | 77.00 | 0.32 | 53.43 | 0.06 | 76.19 | **0.45** |
| Deepseek-R1 | 671B | 91.17 | 0.73 | 76.80 | 0.47 | 71.22 | 0.16 | 67.57 | 0.20 | 76.94 | 0.38 |
| LLaMA-3.2-3B-Instruct (*SFT*) | 3B | 75.25 | 0.48 | 74.24 | 0.40 | 67.67 | 0.16 | 55.14 | 0.09 | 69.63 | 0.29 |
| LLaMA-3.2-3B-Instruct (*Ours*) | 3B | 83.75 | 0.60 | 77.73 | 0.47 | 72.37 | 0.22 | 62.43 | 0.06 | 75.20 | 0.36 |
| Qwen2-7B-Instruct (*SFT*) | 7B | 85.33 | 0.63 | 78.60 | 0.48 | 74.06 | 0.24 | 56.29 | 0.03 | 75.58 | 0.37 |
| Qwen2-7B-Instruct (*Ours*) | 7B | 86.92 | 0.68 | 8.73 | 0.53 | 79.11 | 0.33 | 63.86 | 0.11 | **78.75** | 0.40 |
| Set4 (**200K-250K**) | | | | | | | | | | | |
| LLaMA-3.2-3B-Instruct | 3B | 48.52 | 0.11 | 49.50 | 0.15 | 36.50 | 0.00 | 50.33 | 0.00 | 45.11 | 0.07 |
| Qwen2-7B-Instruct | 7B | 45.19 | 0.00 | 52.50 | 0.15 | 47.67 | 0.00 | 42.00 | 0.00 | 47.07 | 0.03 |
| LLaMA-3.1-8B-Instruct | 8B | 55.00 | 0.15 | 49.25 | 0.15 | 55.50 | 0.07 | 48.67 | 0.07 | 52.88 | 0.11 |
| GPT-4o-mini | 8B | 58.07 | 0.07 | 41.50 | 0.20 | 55.83 | 0.03 | 63.67 | 0.00 | 54.65 | 0.08 |
| Qwen2.5-14B-Instruct | 14B | 55.55 | 0.00 | 59.75 | 0.25 | 51.93 | 0.00 | 65.53 | 0.00 | 56.75 | 0.05 |
| GPT-4o | 200B | 55.19 | 0.22 | 61.25 | 0.25 | 57.50 | 0.03 | 62.33 | 0.07 | 58.42 | **0.14** |
| Deepseek-R1 | 671B | 60.19 | 0.15 | 54.50 | 0.25 | 53.00 | 0.00 | 62.33 | 0.07 | 56.96 | 0.11 |
| LLaMA-3.2-3B-Instruct (*SFT*) | 3B | 54.81 | 0.07 | 60.00 | 0.20 | 63.17 | 0.13 | 47.33 | 0.00 | 57.45 | 0.11 |
| LLaMA-3.2-3B-Instruct (*Ours*) | 3B | 56.89 | 0.04 | 2.25 | 0.20 | 65.17 | 0.10 | 62.00 | 0.07 | 61.59 | 0.10 |
| Qwen2-7B-Instruct (*SFT*) | 7B | 59.26 | 0.07 | 65.25 | 0.30 | 59.83 | 0.10 | 54.00 | 0.00 | 59.89 | 0.12 |
| Qwen2-7B-Instruct (*Ours*) | 7B | 61.48 | 0.15 | 67.75 | 0.30 | 66.67 | 0.07 | 65.33 | 0.00 | **65.16** | 0.13 |

# F  FULL PERFORMANCE OF EFFECTIVENESS ON LOONGFIN

Table 10 presents the full results of the Table 1 in the main manuscript.

