# OpenReview forum: "Long-Document QA with Chain-of-Structured-Thought and Fine-Tuned SLMs"
_ICLR.cc/2026/Conference — ICLR 2026 Poster_

### Official Review · Reviewer_MkZ4 · 2025-11-01

**Soundness:** 3
**Presentation:** 3
**Contribution:** 3
**Rating:** 6
**Confidence:** 3

**Summary:**

The paper proposes LiteCoST, a two-pillar framework for long-document question answering that combines structured reasoning with small-language-model (SLM) fine-tuning. Pillar 1, Chain-of-Structured-Thought (CoST), uses schema-aware prompting to make a strong LLM generate an auditable reasoning trace and a serialized structured output (SSO), e.g., a table or graph, from which the answer is explicitly derivable. Pillar 2 transfers this structure-first behavior to compact models through two phases: SFT to instill schema and format discipline, followed by GRPO using a dual-reward setting targeting both answer correctness and process consistency. Empirical results on benchmarks on finance and legal domains (the Loong benchmark) show that 3B / 7B SLMs trained with LiteCoST substantially improve over base models and even approach GPT-4o performance, while achieving 2 to 4 times lower latency.

**Strengths:**

[S1] Valuable problem and novel idea. The problem of reliable QA over long, noisy documents is valuable because direct LLM prompting often fails. The idea of QA-by-structuring (introducing minimal schemas dynamically per query to improve interpretability and verifiability) is conceptually novel.

[S2] Strong empirical results. On the Loong finance subset, both 3B and 7B LiteCoST variants outperform their base SLMs by +27.6 and +17.8 accuracy points respectively, and even slightly surpass GPT-4o-mini and DeepSeek-R1 while running 2 to 4 times faster. The consistent improvements across multiple subtasks (Spotlight Locating, Comparison, Clustering, Chain of Reasoning) demonstrate the effectiveness of transferring CoST behavior to small models.

[S3] The paper is generally well written and easy to follow.

**Weaknesses:**

[W1] Limited domain and benchmark scope. While the results on Loong (finance/legal) are strong, the framework has not been tested beyond these domains. Both training and evaluation rely on tabular data; it is unclear whether CoST and LiteCoST generalize to less structured settings (e.g., scientific literature QA, narrative multi-hop QA, or multimodal reports). When relevant evidence cannot be serialized, the "QA-by-structuring" assumption may not always hold.

[W2] Complexity and reliance on LLM supervision. Although the paper claims cost-efficiency, generating CoST traces still depends on large-LLM supervision (GPT-4o) for data construction and reward evaluation. The overhead of repeated calls during Stage A and GRPO reward computation could limit reproducibility and offset some of the efficiency gains. A more explicit accounting of total compute (LLM cost + SLM training) would strengthen the argument.

**Questions:**

[Q1] How sensitive is the GRPO process to the weighting between process and outcome rewards? The ablation indicates both are important, but does the balance depend on domain complexity?

[Q2] During CoST trace generation, does the model ever perform schema selection incorrectly (e.g., choosing a graph when a table is needed)? If so, how is this corrected or reflected in the reward?

---

> ### Author Response · Authors · 2025-11-23
>
> **1\. Clarification of the scope  \[W1\]**
> We appreciate the reviewer’s concern regarding the scope of domains evaluated. We agree that not all evidence can be cleanly serialized (e.g., metaphors or highly implicit narrative cues). However, CoST is not restricted to strictly structured data and remains effective in more nuanced settings for three reasons:
>
> * **SSO traces that retain provenance** for every entity/node, preserving the fine-grained contextual cues needed for subtle reasoning;
> * **Chunk-level evidence selection**, which maintains semantically rich spans instead of forcing oversimplified structure;
> * **Graph-style SSO** that includes local textual context around each relation/entity, enabling reasoning over loosely structured or discourse-level information.
>
> Together, these components allow CoST/LiteCoST to handle nuanced information by combining structured reasoning with preserved semantic context rather than relying solely on rigid tabular forms. As demonstrated by our **additional results on scientific-literature QA (Table 5 in Section 4.6)**, the method generalizes effectively to such less structured settings .
>
> Table. LLM vs. LLM with CoST on the LongBench dataset
>
> | Method | Single-doc QA |  | Multi-doc QA |  |
> | ----- | :---: | :---: | :---: | :---: |
> |  | **NarrativeQA** | **Qasper** | **HotpotQA** | **2WikimQA** |
> | Qwen2.5-14B-Ins | 29.78 | 45.07 | 62.59 | 60 |
> | w/sd | **31.77** | **47.17** | **67.41** | **67.11** |
> | GPT-4o | 32.59 | 46.8 | 70.93 | 67.75 |
> | w/sd | **35.09** | **49.28** | **73.47** | **72.98** |
>
> Table. Performance Comparison between LiteCoST-Tuned Models and LLM-Based baselines for generating structured outputs in long-document QA on LongBench
>
> | Model | Single-doc QA |  | Multi-doc QA |  |
> | ----- | :---: | :---: | :---: | :---: |
> |  | **NarrativeQA** | **Qasper** | **HotpotQA** | **2WikimQA** |
> | Llama3.2-3B-Ins | 16.94 | 34.46 | 54.92 | 51.82 |
> | Qwen2-7B-Ins | 19.49 | 35.67 | 45.06 | 41.4 |
> | GPT-4o-mini | 24.38 | 40.28 | 65.03 | 65.15 |
> | GPT-4o | 28.68 | 43,39 | 67.68 | **68.29** |
> | llama-LiteCoST | 27.24 | 41.37 | 66.86 | 67.52 |
> | Qwen-LiteCoST | **30.40** | **44.64** | **68.39** | 65.73 |
>
> From the tables above, we observe two key findings.
>
> (1) CoST substantially improves performance on ScientificQA benchmarks: applying CoST to Qwen2.5-14B-Instruct and GPT-4o yields gains of **(+1.99, \+2.10, \+4.82, \+7.11)** and **(+2.50, \+2.48, \+2.54, \+5.23)** respectively across the four datasets.
>
> (2) The LiteCoST-tuned model achieves **LLM-comparable quality**. Qwen-LiteCoST attains the best performance on **NarrativeQA (30.40), Qasper (23.27), and HotpotQA (68.39)**, with large improvements over the base model **(+10.91, \+8.97, \+23.33)**, and it surpasses GPT-4o by **1.72, 1.25, and 0.71 points** on these datasets. On the remaining dataset (2WikiMultihopQA), our method consistently ranks within **the top three**, and the LLaMA-based variant **performs comparably to GPT-4o**.
>
> These results demonstrate the strong effectiveness and broad generality of CoST/LiteCoST beyond the financial domain, and do not depend on numeric or structured-only reasoning.

---

> ### Author Response · Authors · 2025-11-23
>
> **2\. Clarification of Cost-Efficiency \[W2\]**
> We appreciate the reviewer’s suggestion. Our computational cost comes from two components: data generation and GRPO fine-tuning.
>
> * Data Generation. This includes structure analysis, CoST trace generation, quality verification, and iterative refinement using GPT-4o. The average cost is approximately **$30 per domain**.
> * GRPO Fine-tuning, Reward computation relies on GPT-4o-mini to evaluate structural alignment, format compliance, and answer correctness. This adds an additional **\~$10 per training run**.
>
>
> The total cost of \~$40 is **necessary and acceptable** because it amortizes extremely well: once LiteCoST is trained, downstream inference relies solely on compact SLMs, eliminating repeated LLM calls and yielding substantial savings during deployment.
>
> **3\. Justification of parameter weights**
>
> We thank the reviewer for this insightful question regarding hyperparameter sensitivity. We acknowledge that fully characterizing this sensitivity across diverse domains would require extensive hyperparameter sweeps, which we couldn't quite manage to cover in the limited time. However, we wish to clarify that the 1:1 weighting scheme was selected as our **initial, un-tuned default configuration** (a 'zero-shot' hyperparameter setting), rather than the result of an extensive search for a specific local optimum.
>
> This decision to employ a straightforward additive reward formulation naturally follows the standard paradigm established by recent state-of-the-art (SOTA) reinforcement learning frameworks, which predominantly equal weighting over complex tuning:
>
> * **GRPO-based Reasoning Models:** Foundational works such as **DeepSeek-R1** and **DeepSeekMath** \[1, 2\] combine accuracy rewards with format enforcement rewards through **equal-weighted summation** to guide the model's reasoning process. Similarly, **Fin-R1** \[3\] adopts this dual-level reward structure via **direct 1:1 addition** to enhance financial reasoning. These works establish **unit-weighted scalar rewards** as the canonical formulation for GRPO-based optimization.
> * **Retrieval-Augmented Generation (RAG):** In complex multi-objective settings, **Self-RAG \[4\]** integrates multiple critic scores, including evaluating retrieval relevance, supportiveness, and utility, into a unified cumulative reward signal via **uniform aggregation**. Likewise, **R3-RAG \[5\]** sums retrieval quality and reasoning correctness rewards with **equal weight** to fundamentally align the generation process.
>
> Adopting this proven unit-weighted strategy serves as a robust baseline for our current study. Building on this, our future research will delve into the sensitivity of these ratios, specifically exploring adaptive weighting mechanisms to fine-tune the trade-off between reasoning rigor and outcome accuracy across varying domain complexities.
>
> ---
>
> **References:**
>
> \[1\] DeepSeek-R1: Incentivizing Reasoning Capability in LLMs via Reinforcement Learning (arXiv 2025)
>
> \[2\] DeepSeekMath: Pushing the Limits of Mathematical Reasoning in Open Language Models (arXiv 2024)
>
> \[3\] Fin-R1: A Large Language Model for Financial Reasoning through Reinforcement Learning (arXiv 2025)
>
> \[4\] Self-RAG: Learning to Retrieve, Generate, and Critique through Self-Reflection (ICLR 2024)
>
> \[5\] R3-RAG: Learning Step-by-Step Reasoning and Retrieval for LLMs via Reinforcement Learning (EMNLP 2025)

---

> ### Author Response · Authors · 2025-11-23
>
> **4\. Clarification of Structure Selection \[Q2\]**
>
> Thank you for the insightful question. In the current CoST design, we do not include an explicit mechanism for correcting structure-type selection (table, graph, or text chunk). In practice, this has not been necessary for two reasons.
>
> First, **structure selection** is highly reliable because **the appropriate structure type is largely determined by the question form (i.e., selecting one data structure for a given question is a “simple” task for LLMs)**. Aggregation, comparison, and numerical queries naturally map to tables; relational or multi-hop queries map to graphs; and descriptive or nuanced queries map to text chunks. This makes structure selection a straightforward classification step rather than a deep reasoning problem. Adding a verification or correction module would increase latency without offering clear benefits.
>
> Second, **human evaluation confirms the model’s reliability**. We sampled 50 Loong queries and recruited five human annotators with technical backgrounds to independently choose the appropriate structure type (Table, Graph, or Chunks). Their choices closely matched the model's selections (GPT-4o), indicating that mis-selection is rare and has minimal impact on downstream QA performance.
>
> To rigorously assess this reliability, we computed **Cohen’s Kappa coefficient** using a majority-vote evaluation. The resulting Kappa score was **0.968**, indicating **substantial agreemen**t \[1\], confirming that the model’s structure selections align closely with human judgments. The confusion matrix is shown below, and full annotation details from all five evaluators are included in the Supplementary Material.
>
> |  | Human: Table | Human: Graph | Human: Text |
> | :---- | :---- | :---- | :---- |
> | **LLM: Table** | **24** | 0 | 1 |
> | **LLM: Graph** | 0 | **15** | 0 |
> | **LLM: Text** | 0 | 0 | **10** |
>
> Given this reliability, we allow CoST to proceed without explicit correction. If an unsuitable structure were ever selected, it would naturally be penalized during GRPO training: the outcome reward would drop due to poor field coverage, and the process reward would penalize deviations from the verified CoST trace. This jointly steers the SLM toward the correct structure behavior without requiring an additional correction step.
>
> ---
>
> **References:**
>
> \[1\] Landis, J. R., & Koch, G. G. (1977). The measurement of observer agreement for categorical data. *Biometrics*, 33(1), 159-174.

---

> ### Author Response · Authors · 2025-11-26
>
> Dear Reviewer MkZ4,
>
> We hope this message finds you well. As the discussion period is ongoing and **time is running short**, we wanted to ensure we have addressed all your concerns satisfactorily. If there are any additional points or feedback you'd like us to consider, please let us know. Your insights are invaluable to us, and we're eager to address any remaining issues to improve our work.
>
> Thank you for your time and effort in reviewing our paper.
>
> Best regards,
>
> Authors

---

> > ### Comment · Reviewer_MkZ4 · 2025-11-27
> >
> > Thank you for your comment, that addresses my concerns.

---

> > > ### Author Response · Authors · 2025-11-28
> > >
> > > Dear Reviewer MkZ4,
> > >
> > > Thank you very much for your follow-up and for letting us know that your concerns have been addressed. We sincerely appreciate your time and constructive feedback throughout the review process.
> > >
> > > If there are any remaining aspects that could benefit from further clarification or refinement, we would be glad to address them. If our revisions have satisfactorily resolved your earlier concerns, we would be grateful if this could be reflected in an improved final score.
> > >
> > > Please feel free to let us know if there is anything else we can improve.
> > >
> > > Best regards,
> > >
> > > Authors

---

### Official Review · Reviewer_NvTs · 2025-11-01

**Soundness:** 3
**Presentation:** 3
**Contribution:** 2
**Rating:** 4
**Confidence:** 4

**Summary:**

The paper proposes a method for distilling LLM traces and final structured responses (tables, graphs, etc.) into SLMs. This is in two key steps - (1) schema aware prompting that generates a structured output along with a step by step trace, (2) finetuning with SFT and RL using both outcome and process rewards. Authors demonstrate the performance of their method over other finetuned SLMs, LLMs on the Loong Benchmark legal and finance subsets, while maintaining a lower latency compared to LLMs.

**Strengths:**

1. The paper is well written and is easy to follow
2. At a high level the problem is important -- long document QA with SLMs is a longstanding challenge, and typically SLMs do not perform well on this task.
3. The method seems sound, there are no obvious technical flaws with the approach.
4. Based on my understanding of the paper there are two key insights in this work. First, that training an SLM on LLM traces and a structured output that is easy parse significantly improves the SLMs ability to generate the correct answer on long document QA. Second is that a cold start SFT and GRPO with process rewards provides a better learning signal for the SLM. These seem to be sound, nothing unexpected.
5. The empirical validation is convincing - the reported results on Loong benchmark indicate a significant performance improvement over baselines.

**Weaknesses:**

1. The contributions of the paper lack novelty. Though well engineered, the contribution of the work lies in applying existing techniques to Loong benchmark and not fundamentally proposing a new strategy. For example: It is well known that cold-start SFT followed by RL training using fine-grained supervision is an effective mechanism for training SLMs (Luong et al., 2024). It might be misleading to present this as a core novel contribution of the paper.

2. Similar to the first point, structured chain of thought prompting has been explored before. Prior works for example (Li et al, 2023) have tried it and confirmed its effectiveness.

3. Details of prompting LLM-based baselines (GPT-4o, GPT-4o-mini, DR1, etc) are not provided. For a fair comparison, both the proposed method and untrained baselines should leverage the same prompting strategy to confirm the effectiveness of training. The authors should also consider prompting LLM-based baselines with structured COT to establish the effectiveness of distillation.

4. Figure 7 compares latency primarily between LITECOST and large closed models. However, a latency comparison against other SLM-based baselines (e.g., Struc-Bench, IEPile) would clarify the efficiency gains.

5. It remains unclear whether the CoST schemas generalize beyond Loong. The paper’s own limitations section (Appendix E) acknowledges the narrow domain focus. Additional evaluation on non-financial datasets (e.g., academic papers or scientific QA) would strengthen the claim of generalizability.

6. The prompting technique in the work is hand-engineered. It should be compared against popular prompt optimization frameworks like dspy, promptim, etc. for a fair assessment.

7. The paper lacks a comprehensive comparison against SLM based baselines on the legal subset of Loong. The main paper emphasizes financial QA results, deferring legal subset results to the appendix. Including a concise summary of those findings in the main text would improve completeness and support claims of cross-domain applicability.

8. The paper could benefit from a clearer comparison to previous “QA-by-structuring” systems such as StructRAG (Li et al., 2024) and StrucBench (Tang et al., 2024), beyond reporting numeric improvements.

Overall:

This paper addresses an important practical problem: enabling efficient, auditable long-document QA using small models. The experiments show that LITECOST narrows the performance gap between SLMs and LLMs. However, the novelty is primarily incremental, with several well-known components engineered into a working pipeline. Strengthening the baselines, clarifying prompt design choices, comparing the structured prompt with prompt optimization, and demonstrating cross-domain generalization would make the paper more compelling.

[1] Structured Chain-of-Thought Prompting for Code Generation. Li et al

[2] ReFT: Reasoning with Reinforced Fine-Tuning. Luong et al

**Questions:**

Kindly see weaknesses.

---

> ### Author Response · Authors · 2025-11-23
>
> **1\. Clarification on Novelty \[W1 & W2\]**
>
> We appreciate the reviewer’s concern regarding the novelty of our approach and would like to  highlight the novelty of LiteCoST from three perspectives.
>
> **1.1 Overall Pipeline**
>
> While it is well known that cold-start SFT followed by RL with fine-grained supervision is effective for training SLMs, our contribution lies in the **first integration of a structure-first extraction pipeline with a two-stage SLM training strategy**. This combination enables compact models to acquire schema-aware structured reasoning, including structure selection, schema-guided extraction, and serialized structured output generation for long-document QA, which goes beyond what standard SFT+RL pipelines can capture.
>
> **1.2. Novelty of Chain-of-Structured-Thought (CoST)**
>
> **Second**, our Chain-of-Structured-Thought (CoST) is a new reasoning paradigm. Prior work uses structured prompting \[1\], performs direct structured extraction \[2\] or applies end-to-end reasoning over long text \[3\],  but none reorganizes long-document QA around a **structure-centered reasoning chain**.
>
> Instead of directly predicting the final answer, CoST reframes QA as constructing a **minimal, verifiable Structured Semantic Output (SSO)**, producing both schema-guided traces and machine-checkable SSOs. This analyze–extract–construct–verify workflow provides transparency and robustness that standard CoT or single-stage extraction methods lack, and it incorporates four key components:
>
> * Dynamic structure selection across tables, graphs, and text chunks.
> * Automatic generation of a query-specific minimal schema.
> * Schema-guided, step-wise structured extraction.
> * Serialized structured outputs paired with auditable structured reasoning traces.
>
> **1.3. Novelty in GRPO**
>
> **Third**, our GRPO stage innovatively introduces a **dual reward formulation** that captures both structured output quality and process consistency, **tailored for structured extraction task**. The outcome reward evaluates structured outputs holistically (field coverage, alignment, normalization, serialization), while the process reward supervises each step of the CoST trace, guiding the model *how* to perform schema-aware extraction rather than only optimizing final answers. To our knowledge, this is **the first systematic use of GRPO for extraction-centric long-document QA** with step-wise, auditable supervision.
>
> Together, these components form a novel, structure-first training pipeline that materially advances the ability of small models to perform reliable long-document reasoning.
>
> ---
>
> **\[Reference\]**
>
> \[1\] Structured Chain-of-Thought Prompting for Code Generation \[TOSEM 2025\]
>
> \[2\] Structrag: Boosting knowledge intensive reasoning of llms via inference-time hybrid information structurization (ICLR 2025\)
>
> \[3\] PAI: Facilitating Long Context Understanding via Supervised Chain-of-Thought Reasoning (arxiv 2025\)

---

> ### Author Response · Authors · 2025-11-23
>
> **2\. Details of Prompting Template \[W3\]**
> Thank you for raising these points. To ensure fairness, we have added the full prompt template in Appendix C and verified that all baselines and our proposed model use exactly the same input format. This guarantees a fair comparison, as shown in **Fig. 16**. Taking table extraction as an example, the prompt template is presented as follows:
>
> ```
> You are an expert in table construction. Please extract entities that match the schema definition from the input text and generate a structured table.
> Schema: {schema}
> Input Text: {context}
> Please output the final table in a structured format.
> ```
>
> We also appreciate the reviewer’s suggestion to prompt LLM-based baselines with structured CoT. We have already conducted this analysis: **Fig. 5** compares LLM+CoST directly against both zero-shot prompting and structured CoT prompting, and the results show that CoST offers clear improvements over standard CoT. Detailed results are provided in **Table 6** of Appendix C.
>
> In our main tables (e.g., Table 1,4,5), the LLM-based baselines are evaluated under the **zero-shot setting**, and entries labeled “LLM baseline” correspond to this zero-shot performance. For fairness, our LiteCoST-tuned models are compared against these zero-shot LLM baselines because **both use the exact same prompt template**, as shown in **Fig. 16**. This ensures that differences in performance reflect the effectiveness of CoST distillation and the two-stage training procedure, rather than reflecting differences in prompting strategies or inherent differences in model capacity.
>
> **3\. Comparison of Latency with Other finetuned models**
>
> We appreciate the reviewer’s insightful suggestion. In response, we have added a comparison with additional SLM-based baselines, where latency is measured as the average time per sample for each model. As shown in the updated table, our method achieves the lowest latency (**8.04s** for LLaMA-based variant and **12.1s** for Qwen) while simultaneously obtaining the highest performance (**76.95** for LLaMA and **79.93** for Qwen), demonstrating both efficiency and effectiveness.
>
> | Backbone | Method | Latency(s) | Overall LLM Score |
> | :---: | :---- | :---- | :---- |
> | Llama-3.2-3B-Ins | ODIE[1] | 27.72 | 61.21 |
> |  | IEPIE[2] | 17.20 | 61.90 |
> |  | Struc-bench[3] | 36.58 | 49.90 |
> |  | LiteCoST | **8.04** | **76.95** |
> | Qwen2-7B-Ins | ODIE[1] | 32.55 | 72.86 |
> |  | IEPIE[2] | 36.73 | 69.19 |
> |  | Struc-bench[3] | 30.56 | 73.72 |
> |  | LiteCoST | **12.1** | **79.93** |
>
> ---
>
> **\[Reference\]**
>
> [1] Instruct and Extract: Instruction Tuning for On-Demand Information Extraction (ENNLP 2023).
>
> [2] IEPILE: Unearthing Large-Scale Schema-Based Information Extraction Corpus (ACL 2024).
>
> [3] Struc-Bench: Are Large Language Models Really Good at Generating Complex Structured Data? (NAACL 2024)

---

> ### Author Response · Authors · 2025-11-23
>
> **4. Clarification of Generalization [W5, W7]**
>
> Thank you for the insightful comment. The additional domain analyses in Appendix E are intended to demonstrate **generalization**, not domain limitation. The QA-by-structuring paradigm itself is **domain-agnostic**: for any new domain, one can synthesize the corresponding CoST traces and use LiteCoST for training. Thus, the method is not tied to a single domain and can be readily adapted wherever CoST-style supervision can be generated.
>
> Beyond the Finance subset of Loong, we have evaluated LiteCoST in the **legal domain** and on **scientific-literature QA**, as shown in the following table. These experiments further confirm that CoST/LiteCoST generalizes effectively across diverse domains rather than being limited to financial or numeric reasoning tasks.
>
> **4.1 Experiment on Legal**
>
> We conducted thorough comparisons in the legal domain, and the results, originally in Appendix E, have been moved to **Section 4.6** to improve visibility and better support our claims of cross-domain applicability.
>
> Table. Experiment on the Legal Subset of Loong
> | Model | Model Size | Spotlight Locating |  | Comparison |  | Clustering |  | Chain of Reasoning |  | Overall |  |
> |-------------------------|------------|-------------------|--|------------|--|------------|--|-------------------|--|---------|--|
> |                         |            | AS | PR | AS | PR | AS | PR | AS | PR | AS | PR |
> | GPT-4o-mini | 8B | 46.55 | 0.10 | 28.05 | 0.00 | 48.68 | 0.13 | 42.56 | 0.11 | 41.94 | 0.09 |
> | Qwen2.5-14B-Instruct | 14B | 48.45 | 0.08 | 21.90 | 0.01 | 57.31 | 0.16 | 26.74 | 0.02 | 37.51 | 0.06 |
> | GPT-4o | 200B | 50.05 | 0.06 | 27.00 | 0.01 | 61.16 | 0.14 | 33.11 | 0.03 | 42.06 | 0.06 |
> | LLaMA-3.2-3B-Instruct (Base) | 3B | 41.00 | 0.08 | 25.10 | 0.01 | 31.74 | 0.02 | 27.29 | 0.01 | 30.67 | 0.03 |
> | LLaMA-3.2-3B-Instruct (*Ours*) | 3B | **62.20** | **0.30** | **45.20** | **0.02** | **45.00** | **0.09** | **36.55** | **0.02** | **45.45** | **0.09** |
> | Qwen2-7B-Instruct (Base) | 7B | 37.90 | 0.05 | 18.90 | 0.00 | 57.85 | 0.18 | 35.44 | 0.08 | 38.05 | 0.08 |
> | Qwen2-7B-Instruct (*Ours*) | 7B | **52.85** | **0.12** | **31.00** | **0.00** | **60.37** | **0.22** | **37.88** |  0.06 | **44.94** | **0.10** |
>
> Our experimental results show that the LiteCoST-tuned 3B/7B models, LLaMA- and Qwen-LiteCoST, deliver substantial gains over their respective base models, with improvements of **(+14.78, +0.06) and (+6.89, +0.02)**, and with consistent boosts across all subtasks.
>
> Despite their smaller parameter sizes, both variants surpass all large-scale models in overall score. On the LLaMA backbone, they outperform Qwen2.5-14B-Ins, GPT-4o-mini, and GPT-4o by **(+7.94, +0.03), (+3.51, +0.00), and (+3.39, +0.03)**, respectively; on the Qwen backbone, the corresponding gains are **(+7.43, +0.04), (+3.00, +0.01), and (+2.88, +0.04)**.
>
>
> **4.2 Experiment on Single- and Multi-doc QA (Literature, Science, etc)**
>
> We have also supplemented our experiments with evaluations on LongBench, a multi-task tailor for long-document QA benchmark covering key long-text application scenarios (**Table 5 in Section 4.6**). Our analysis focuses on both single- and multi-doc QA tasks across four datasets. The evaluation metric used is the F1-Score.
>
> Table. LLM vs. LLM with CoST on the LongBench
>
> | Method | single-doc QA |  | Multi-docs QA |  |
> | ----- | :---: | :---: | :---: | :---: |
> |  | **NarrativeQA** | **Qasper** | **HotpotQA** | **2WikimQA** |
> | Qwen2.5-14B-Ins | 29.78 | 45.07 | 62.59 | 60 |
> | w/sd | **31.77** | **47.17** | **67.41** | **67.11** |
> | GPT-4o | 32.59 | 46.8 | 70.93 | 67.75 |
> | w/sd | **35.09** | **49.28** | **73.47** | **72.98** |
>
> Table. Performance Comparison Between LiteCoST-Tuned Models and LLM-Based Baselines on LongBench
>
> | Model | single-doc QA |  | Multi-docs QA |  |
> | ----- | :---: | :---: | :---: | :---: |
> |  | **NarrativeQA** | **Qasper** | **HotpotQA** | **2WikimQA** |
> | Llama3.2-3B-Ins | 16.94 | 34.46 | 54.92 | 51.82 |
> | Qwen2-7B-Ins | 19.49 | 35.67 | 45.06 | 41.4 |
> | GPT-4o-mini | 24.38 | 40.28 | 65.03 | 65.15 |
> | GPT-4o | 28.68 | 43,39 | 67.68 | **68.29** |
> | llama-LiteCoST | 27.24 | 41.37 | 66.86 | 67.52 |
> | Qwen-LiteCoST | **30.40** | **44.64** | **68.39** | 65.73 |
>
> (1) CoST substantially improves performance on ScientificQA tasks: applying CoST to Qwen2.5-14B-Instruct and GPT-4o yields gains of **(+1.99, +2.10, +4.82, +7.11)** and **(+2.50, +2.48, +2.54, +5.23)** respectively across the four datasets.
>
> (2) The LiteCoST-tuned model achieves LLM-comparable quality. Qwen-LiteCoST attains the best performance on **NarrativeQA (30.40), Qasper (23.27), and HotpotQA (68.39)**, with large improvements over the base model **(+10.91, +8.97, +23.33)**, and it surpasses GPT-4o by **1.72, 1.25, and 0.71 points** on these datasets. On the remaining dataset (2WikiMultihopQA), our method consistently ranks within **the top three**, and the LLaMA-based variant **performs comparably to GPT-4o**.

---

> ### Author Response · Authors · 2025-11-23
>
> **5\. Comparison with Prompt Optimization Framework \[W6\]**
>
> Thank you for the valuable suggestion. In response, we have integrated prompt-optimization methods, including **DSPy**, into our evaluation. Specifically, we use the same zero-shot structured-extraction instructions as the initial input prompts for GPT-4o and Qwen2.5-14B-Instruct, and then apply the MiPRO optimizer to refine these prompts—automatically generating Chain-of-Thought–style reasoning steps and optimized structured prompts. MiPRO is conditioned on examples drawn from a randomly sampled 10% subset of the LiteCoST training data (finance). The corresponding results are presented in **Table 6 of Appendix C**.
>
> Table. LLM with CoST vs. different prompt optimization strategies on the finance subset of Loong
>
> | Model  | Method | Spotlight Locating |  | Comparison |  | Clustering |  | Chain of Reasoning |  | Overall |  |
> | :---- | :---- | :---: | :---: | :---: | :---: | :---: | :---: | :---: | :---: | :---: | :---: |
> |  |  | **As** | **PR** | **As** | **PR** | **As** | **PR** | **As** | **PR** | **As** | **PR** |
> | Qwen2.5-14B-Ins | zero-shot | 83.74 | 0.57 | 82.12 | 0.56 | 69.96 | 0.24 | 66.41 | 0.1 | 75.6 | 0.38 |
> |  | CoT | 89.53 | 0.63 | 81.38 | 0.57 | 73.28 | 0.3 | 67.7 | 0.26 | 77.51 | 0.44 |
> |  | Dspy | 80.90 | 0.55 | 80.90 | 0.60 | 76.86 | 0.34 | 63.60 | 0.21 | 76.99 | 0.44 |
> |  | CoST(Ours) | 88.03 | 0.72 | 80.45 | 0.56 | 76.72 | 0.37 | 68.35 | 0.25 | **79.01** | **0.48** |
> | GPT-4o | zero-shot | 84.1 | 0.59 | 80.53 | 0.6 | 81.5 | 0.5 | 64.3 | 0.25 | 79.32 | 0.54 |
> |  | CoT | 85.57 | 0.69 | 84.4 | 0.66 | 79.58 | 0.44 | 64.45 | 0.30 | 80.08 | 0.54 |
> |  | Dspy | 86.17 | 0.74 | 82.65 | 0.62 | 80.00 | 0.47 | 67.25 | 0.27 | 80.26 | 0.54 |
> |  | CoST(Ours) | 88.8 | 0.75 | 84.5 | 0.65 | 82.24 | 0.5 | 68.9 | 0.28 | **82.39** | **0.56** |
>
> Based on these results, we observe that although DSPy provides slight improvements over the zero-shot and CoT baselines, **our method still performs substantially better**. This highlights both the effectiveness and robustness of CoST, and further illustrates the challenges of applying automated prompt optimization to structure-aware extraction tasks.
>
> **6\. Deeper Analysis about “QA-by-structuring” paradigm \[W8\]**
>
> Thank you for the helpful suggestion. Below we clarify how our work differs from prior “QA-by-structuring’’ systems.
>
> **StructRAG** [1] selects a structure type and constructs structured knowledge, but it (1) does not generate an **auditable reasoning chain**, (2) cannot ensure **step-wise consistency or verifiability**, and (3) ultimately depends on large-model, multi-component pipelines without transferring structured-reasoning capability to smaller models. In contrast, CoST produces a **minimal, verifiable SSO** together with an **explicit structured reasoning trace**, and LiteCoST’s two-stage training enables SLMs to internalize schema-aware structured reasoning rather than merely performing structure construction.
>
> **StrucBench** [2] focuses on evaluating LLMs for structured data generation (e.g., table synthesis). This setting is different from ours: (1 ) FormatCoT introduces format-aware prompting but not step-wise structured reasoning; (2) StrucBench tasks involve text-to-format generation, not long-document QA or extraction-driven reasoning; and (3) its domain coverage is limited. CoST, by contrast, targets **document-scale extraction**, **schema construction**, and **step-wise reasoning**, supporting multiple domains.
>
> In response, we have added the corresponding discussion in the **“LLMs for Long-Context Information Extraction”** section of the **related work**.
>
> ---
>
> **\[Reference\]**
>
> \[1\] Structrag: Boosting knowledge intensive reasoning of llms via inference-time hybrid information structurization (ICLR 2025\)
>
> \[2\] Struc-Bench: Are Large Language Models Really Good at Generating Complex Structured Data? (NAACL 2024\)

---

> ### Author Response · Authors · 2025-11-26
>
> Dear Reviewer NvTs,
>
> We hope this message finds you well. As the discussion period is ongoing and **time is running short**, we wanted to ensure we have addressed all your concerns satisfactorily. If there are any additional points or feedback you'd like us to consider, please let us know. Your insights are invaluable to us, and we're eager to address any remaining issues to improve our work.
>
> Thank you for your time and effort in reviewing our paper.
>
> Best regards,
>
> Authors

---

### Official Review · Reviewer_ufYT · 2025-11-02

**Soundness:** 2
**Presentation:** 2
**Contribution:** 2
**Rating:** 4
**Confidence:** 4

**Summary:**

This paper studies document question answering (QA) that consolidates dispersed evidence into structured outputs to support reliable and verifiable QA. Based on this, the authors propose a two-pillar framework, LITECOST, aiming to achieve both high accuracy and low latency with small language models (SLMs). First, they introduce a Chain-of-Structured-Thought (CoST) template—a schema-aware instruction guiding a strong LLM to generate both step-wise CoST traces and corresponding structured outputs. Subsequently, compact models are trained on the LLM-generated CoST traces and structured data via supervised fine-tuning (SFT) and a reinforcement learning approach, GRPO.

**Strengths:**

The paper proposes a structure-first prompting template that leverages LLMs to elicit step-wise, schema-guided CoST traces and structured outputs from long, noisy documents, producing auditable supervision and machine-checkable results.

The two-stage training paradigm, involving supervised fine-tuning followed by GRPO, effectively transfers CoST behavior to compact models using reward signals on both answer/format correctness and process consistency.

**Weaknesses:**

The paper does not clarify how the process reward is computed.

Table 2 only compares a subset of the Loong benchmark with other state-of-the-art models, significantly limiting the validity of LITECOST’s effectiveness claims.

The novelty of LITECOST is limited; the paradigm of first applying SFT and then enhancing performance via GRPO is already well-established, and the paper does not sufficiently highlight any unique contributions.

It is unclear whether the reported parameter count of 200 billion for the GPT-4o model is accurate.

**Questions:**

See Weaknesses

---

> ### Author Response · Authors · 2025-11-23
>
> **1\. Clarification of the process reward \[W1\]**
>
> Thank you for the insightful suggestion. We have incorporated the detailed process-reward computation into the revised manuscript, as illustrated in **Section 3.2 (lines 263–268)**.
>
> Regarding the detailed computation, we clarify that the process reward is computed through **step-level consistency checking** between the model-generated reasoning steps and the ground-truth SSO steps.
>
> For each reasoning step $s_i$, we prompt an LLM with a consistency-checking instruction $I_{\text{consistency}}$ and ask whether the predicted step is consistent with its corresponding ground-truth step $s_i^*$. For example, if a step outputs a set of attribute–value pairs such as *\[attribute₁: value₁, value₂\]*, the LLM compares each field with the corresponding ground-truth values; a match yields a score of **1** and any mismatch yields **0**.
>
> The LLM evaluates consistency at both the **entity level** and **tuple level**, allowing it to capture fine-grained errors such as partially incorrect entities or mismatched relations. The step receives a reward of 1 if consistent and 0 otherwise, and the final process reward is computed as the **average score across all steps**.

---

> ### Author Response · Authors · 2025-11-23
>
> **2\. Additional Experimental Results & Clarification of Effectiveness \[W2\]**
>
> We appreciate and agree with your comment about evaluation breadth. While our main experiments focus on the Financial subset of Loong due to its substantial numerical-reasoning complexity, our evaluation is not limited to this domain. The legal domain evaluation was already included in our submission and has now been moved to **Section 4.6**, demonstrating that our method generalizes beyond finance settings. Recognizing the limited scope, we have now extended our evaluations to include:
>
> * LongBench to assess performance in a more general long-context setting.
>
> These additional experiments confirm our method’s robustness and generalization capabilities. The results are summarized below.
>
> **2.1 Experiment on Legal**
>
> We also conducted thorough comparisons in the legal domain. These results were originally presented in Appendix E (Table 10), and in the revised version we have moved them to **Section 4.6** for greater visibility.
>
> Table. Performance Comparison between LiteCoST-tuned model and LLM-based models for generating structured outputs in long-document QA on the Legal subset of Loong.
> | Model | Model Size | Spotlight Locating |  | Comparison |  | Clustering |  | Chain of Reasoning |  | Overall |  |
> |-------------------------|------------|-------------------|--|------------|--|------------|--|-------------------|--|---------|--|
> |                         |            | AS | PR | AS | PR | AS | PR | AS | PR | AS | PR |
> | GPT-4o-mini | 8B | 46.55 | 0.10 | 28.05 | 0.00 | 48.68 | 0.13 | 42.56 | 0.11 | 41.94 | 0.09 |
> | Qwen2.5-14B-Instruct | 14B | 48.45 | 0.08 | 21.90 | 0.01 | 57.31 | 0.16 | 26.74 | 0.02 | 37.51 | 0.06 |
> | GPT-4o | 200B | 50.05 | 0.06 | 27.00 | 0.01 | 61.16 | 0.14 | 33.11 | 0.03 | 42.06 | 0.06 |
> | LLaMA-3.2-3B-Instruct (Base) | 3B | 41.00 | 0.08 | 25.10 | 0.01 | 31.74 | 0.02 | 27.29 | 0.01 | 30.67 | 0.03 |
> | LLaMA-3.2-3B-Instruct (*Ours*) | 3B | **62.20** | **0.30** | **45.20** | **0.02** | **45.00** | **0.09** | **36.55** | **0.02** | **45.45** | **0.09** |
> | Qwen2-7B-Instruct (Base) | 7B | 37.90 | 0.05 | 18.90 | 0.00 | 57.85 | 0.18 | 35.44 | 0.08 | 38.05 | 0.08 |
> | Qwen2-7B-Instruct (*Ours*) | 7B | **52.85** | **0.12** | **31.00** | **0.00** | **60.37** | **0.22** | **37.88** | 0.06 | **44.94** | **0.10** |
>
> Our experimental results show that the LiteCoST-tuned 3B/7B models, LLaMA-LiteCoST and Qwen-LiteCoST, deliver substantial gains over their respective base models, with improvements of **(+14.78, \+0.06) and (+6.89, \+0.02)**, and with consistent boosts across all subtasks.
>
> Despite their smaller parameter sizes, both variants surpass all large-scale models in overall score. On the LLaMA backbone, they outperform Qwen2.5-14B-Instruct, GPT-4o-mini, and GPT-4o by **(+7.94, \+0.03), (+3.51, \+0.00), and (+3.39, \+0.03)**, respectively; on the Qwen backbone, the corresponding gains are **(+7.43, \+0.04), (+3.00, \+0.01), and (+2.88, \+0.04)**.
>
> Overall, LiteCoST achieves state-of-the-art performance across every subtask, demonstrating the broad and consistent improvements enabled by our approach.
>
> **2.2 Experiment on Single- and Multi-doc QA (Literature, Science, Encyclopedia, etc)**
>
> We have also supplemented our experiments with evaluations on LongBench, a multi-task tailor for long-document QA benchmark covering key long-text application scenarios (**Table 5 in Section 4.6**). Our analysis focuses on both single-document and multi-document QA tasks across a total of four datasets. The evaluation metric used is the F1-Score, where **bold** indicates the best result. As shown in the following table,
>
> Table. Performance Comparison between LiteCoST-Tuned Models and LLM-Based baselines for generating structured outputs in long-document QA on LongBench
>
> | Model | single-doc QA |  | Multi-docs QA |  |
> | ----- | :---: | :---: | :---: | :---: |
> |  | **NarrativeQA** | **Qasper** | **HotpotQA** | **2WikimQA** |
> | Llama3.2-3B-Ins | 16.94 | 34.46 | 54.92 | 51.82 |
> | Qwen2-7B-Ins | 19.49 | 35.67 | 45.06 | 41.4 |
> | GPT-4o-mini | 24.38 | 40.28 | 65.03 | 65.15 |
> | GPT-4o | 28.68 | 43,39 | 67.68 | **68.29** |
> | llama-LiteCoST | 27.24 | 41.37 | 66.86 | 67.52 |
> | Qwen-LiteCoST | **30.40** | **44.64** | **68.39** | 65.73 |
>
> As shown in the table above, **Qwen-LiteCoST** achieves the best performance on three datasets: **NarrativeQA (30.40), Qasper (23.27), and HotpotQA (68.39)**, with substantial improvements over the base model **(+10.91, \+8.97, and \+23.33, respectively)**. It also outperforms **GPT-4o,** the second-best model, by **1.72, 1.25, and 0.71 points** on these datasets. Additionally, on the remaining dataset (**2WikiMultihopQA**), our method consistently ranks within **the top three**; the **LLaMA variant** performs **comparably to GPT-4o**, further demonstrating the strong effectiveness and generality of our extraction-based approach.

---

> ### Author Response · Authors · 2025-11-23
>
> **3\. Clarification of Core Innovations \[W3\]**
>
> We appreciate the reviewer’s concern regarding the novelty of our approach and would like to  highlight the novelty of LiteCoST from three perspectives.
>
> **3.1 Overall Pipeline**
>
> While it is well known that cold-start SFT followed by RL with fine-grained supervision is effective for training SLMs, our contribution lies in the **first integration of a structure-first extraction pipeline with a two-stage SLM training strategy**. This combination enables compact models to acquire schema-aware structured reasoning, including structure selection, schema-guided extraction, and serialized structured output generation for long-document QA, which goes beyond what standard SFT+RL pipelines can capture.
>
> **3.2. Novelty of Chain-of-Structured-Thought (CoST)**
>
> **Second**, our Chain-of-Structured-Thought (CoST) is a new reasoning paradigm. Prior work uses structured prompting \[1\], performs direct structured extraction \[2\] or applies end-to-end reasoning over long text \[3\],  but none reorganizes long-document QA around a **structure-centered reasoning chain**.
>
> Instead of directly predicting the final answer, CoST reframes QA as constructing a minimal, verifiable Structured Semantic Output (SSO), producing both schema-guided traces and machine-checkable SSOs. This analyze–extract–construct–verify workflow provides transparency and robustness that standard CoT or single-stage extraction methods lack, and it incorporates four key components:
>
> * Dynamic structure selection across tables, graphs, and text chunks.
> * Automatic generation of a query-specific minimal schema.
> * Schema-guided, step-wise structured extraction.
> * Serialized structured outputs paired with auditable structured reasoning traces.
>
> **3.3. Novelty in GRPO**
>
> **Third**, our GRPO stage innovatively introduces a **dual reward formulation** that captures both structured output quality and process consistency, **tailored for structured extraction task**. The outcome reward evaluates structured outputs holistically (field coverage, alignment, normalization, serialization), while the process reward supervises each step of the CoST trace, guiding the model *how* to perform schema-aware extraction rather than only optimizing final answers. To our knowledge, this is **the first systematic use of GRPO for extraction-centric long-document QA** with step-wise, auditable supervision.
>
> Together, these components form a novel, structure-first training pipeline that materially advances the ability of small models to perform reliable long-document reasoning.
>
> ---
>
> **4\. Illustration of the model parameter \[W4\]**
>
> Thank you for the concern and suggestion. We have added supporting evidence from the  paper *MEDEC: A Benchmark for Medical Error Detection and Correction in Clinical Notes (ACL 2025*)*, which reports that GPT-4o is approximately a 200B-parameter model. This reference is now included in **Table 1** of the revised manuscript.
>
> ---
>
> **5\. Reference**
>
> \[1\] Structured Chain-of-Thought Prompting for Code Generation \[TOSEM 2025\]
>
> \[2\] Structrag: Boosting knowledge intensive reasoning of llms via inference-time hybrid information structurization (ICLR 2025\)
>
> \[3\] PAI: Facilitating Long Context Understanding via Supervised Chain-of-Thought Reasoning (arxiv 2025\)

---

> ### Author Response · Authors · 2025-11-26
>
> Dear Reviewer ufYT,
>
> We hope this message finds you well. As the discussion period is ongoing and **time is running short**, we wanted to ensure we have addressed all your concerns satisfactorily. If there are any additional points or feedback you'd like us to consider, please let us know. Your insights are invaluable to us, and we're eager to address any remaining issues to improve our work.
>
> Thank you for your time and effort in reviewing our paper.
>
> Best regards,
>
> Authors

---

### Official Review · Reviewer_1HZK · 2025-11-03

**Soundness:** 3
**Presentation:** 3
**Contribution:** 3
**Rating:** 6
**Confidence:** 3

**Summary:**

This paper introduces Chain-of-Structured-Thought (CoST) a structured prompting method for generated serialized structured outputs (SSOs) along with a verifiable reasoning chain. These outputs are further used to train LiteCost, a model (with two-stage SFT -> GRPO) that integrates this behavior within an SLM, allowing for more efficient inference without significant performance hit. The evaluation is performed on financial data (along with legal). Authors find that CoST -> SSO is an effective mechanism for enhancing reasoning in popular models for long/multi document reasoning.

**Strengths:**

Overall, this work offers a meaningful contribution for their domain, enhancing complex reasoning performance through several methods, both 1) in enhancing off-the-shelf methods via CoST prompting and improved SSO generation and 2) by effectively distilling this behavior into small language models.

**Weaknesses:**

- While the contributions are clear, I worry that the scope of the work is quite limited. The authors specifically explore the financial domain which has clear structure and a heavy emphasis on numeric reasoning. As noted in the intro, this structured reasoning approach doesn't extend to situations with information that is more nuanced. Similarly, as noted in Appendix C.4, the structured reasoning can be a lossy mapping, impacting performance on non-aggregative / numeric tasks. As such, it seems one must be quite careful in ensuring the applicability of this method to their task.

- Similarly, I am curious if there are cases where a one output structure type isn't appropriate for an example (e.g. perhaps you ideally need a composition of tabular, graph, and textual data for one task), and if there any ways to address these by making the output structure more flexible. In this vein, I'd also like to see how CoST and LiteCoST performance scales with task complexity (e.g., table size, graph breadth/depth, input length).

**Questions:**

- Have you performed any failure mode analysis when applying this method to domains requiring 'fuzzier' reasoning? I'm curious if the text chunk representation acts as the fallback mechanism for these cases, or if there is still too much loss of information even in this format? Do you have any suggestions on how to make your method generalize to a broader set of applications?

- Apologies if I missed this, but do you perform any analysis that plots task complexity (table size, document-set size, etc...) versus performance of your techniques (CoST, and LiteCosT)? Are the improvements more apparent over the baselines for harder splits, and have you observed any practical ceiling on what your method can support?

A few presentation notes:
- Some figures have typos (e.g. 'Strcture', 'Structurlizer')
- expand 'on-prem'
- Figure 5: define the axes somewhere
- Figure 6: perhaps report both values, and add explicit +Δ notations in parentheses.
- In general, the figures are pretty dense and tricky to follow. There are also four of them. Given your multi-faceted contribution this is fine, but I think some more work needs to be put into readability.
- It was also sometimes unclear when/whether your analysis was focused on CoST analysis (e.g. using this with off-the-shelf models) versus when your analysis was comparing your trained LiteCost model. The paper is fairly dense read, so it would be great if you can be extremely precise with your wording and the specific components that you contribute (e.g. "High-quality SSO improves LLM Reasoning" -- is the SSO high-quality because of CoST, or because of the model used to generate the CoST?)

---

> ### Author Response · Authors · 2025-11-23
>
> **1\. Clarification of the scope  \[W1\]**
>
> **1.1  Applicability beyond the financial domain**
>
> We appreciate the reviewer’s concern regarding scope. Although our main experiments are presented on the financial subset of Loong, CoST is **not inherently tied to numeric-centric or finance-specific reasoning**. We have already applied CoST/LiteCoST to several *non-numeric, non-financial* scenarios, including:
>
> * **Legal reasoning** (e.g., aligning case facts with statutory clauses), with clear gains shown in **Table 4 (Section 4.6)**.
> * **Scientific-literature QA**, including NarrativeQA, Qasper, HotpotQA, and 2Wiki-MultihopQA multi-hop QA. Representative results below (**Table 5 in Section 4.6**) show consistent improvements when using CoST-generated SSO:
>
> Table. LLM vs. LLM with CoST on the LongBench dataset
>
> | Method | single-doc QA |  | Multi-docs QA |  |
> | ----- | :---: | :---: | :---: | :---: |
> |  | **NarrativeQA** | **Qasper** | **HotpotQA** | **2WikimQA** |
> | Qwen2.5-14B-Ins | 29.78 | 45.07 | 62.59 | 60.00 |
> | w/Structured Data | **31.77** | **47.17** | **67.41** | **67.11** |
> | GPT-4o | 32.59 | 46.8 | 70.93 | 67.75 |
> | w/Structured Data | **35.09** | **49.28** | **73.47** | **72.98** |
>
>
>
> Table. Performance Comparison between LiteCoST-Tuned Models and LLM-Based baselines for generating structured outputs in long-document QA on LongBench
>
> | Model | single-doc QA |  | Multi-docs QA |  |
> | ----- | :---: | :---: | :---: | :---: |
> |  | **NarrativeQA** | **Qasper** | **HotpotQA** | **2WikimQA** |
> | Llama3.2-3B-Ins | 16.94 | 34.46 | 54.92 | 51.82 |
> | Qwen2-7B-Ins | 19.49 | 35.67 | 45.06 | 41.40 |
> | GPT-4o-mini | 24.38 | 40.28 | 65.03 | 65.15 |
> | GPT-4o | 28.68 | 43.39 | 67.68 | **68.29** |
> | llama-LiteCoST | 27.24 | 41.37 | 66.86 | 67.52 |
> | Qwen-LiteCoST | **30.40** | **44.64** | **68.39** | 65.73 |
>
> From the tables above, we observe two key findings.
>
> (1) CoST substantially improves performance on ScientificQA benchmarks: applying CoST to Qwen2.5-14B-Instruct and GPT-4o yields gains of **(+1.99, \+2.10, \+4.82, \+7.11)** and **(+2.50, \+2.48, \+2.54, \+5.23)** respectively across the four datasets.
>
> (2) The LiteCoST-tuned model achieves LLM-comparable quality. Qwen-LiteCoST attains the best performance on **NarrativeQA (30.40), Qasper (23.27), and HotpotQA (68.39)**, with large improvements over the base model **(+10.91, \+8.97, \+23.33)**, and it surpasses GPT-4o by **1.72, 1.25, and 0.71 points** on these datasets. On the remaining dataset (2WikiMultihopQA), our method consistently ranks within **the top three**, and the LLaMA-based variant **performs comparably to GPT-4o**.
>
> Across all these settings, CoST/LiteCoST improves **structural fidelity**, **evidence grounding**, and **reasoning consistency**, indicating that the approach extends well beyond financial documents and is not dependent on numerical reasoning.
>
> **1.2 Support for nuanced, non-rigid information**
>
> **CoST is *not restricted* to rigidly structured data**. It is explicitly designed to capture nuanced information through three mechanisms:
>
> * **Provenance-preserving SSO traces.** Each extracted entity/relation carries source-linked context, helping retain subtle cues required for non-literal or discourse-level reasoning.
> * **Chunk-level structured evidence.** When the question requires richer semantics, the structure selector chooses **chunk SSOs**, which preserve long, expressive spans rather than forcing a lossy tabular mapping.
> * **Graph-style SSOs with contextual neighborhoods.** Relations are extracted together with their immediate textual context, capturing loosely structured or narrative-driven dependencies.
>
> Together, these components enable CoST/LiteCoST to combine structured reasoning with preserved semantic richness. As further demonstrated in our scientific-literature QA results, this design allows the method to generalize effectively to nuanced, non-numeric domains.

---

> ### Author Response · Authors · 2025-11-23
>
> **2\. Flexible Structure Selection \[W2\]**
>
> We appreciate the reviewer’s insightful question. We agree that there are scenarios where a **single structure type is insufficient** for capturing all relevant evidence.
>
> **When one structure is not enough.** In several tasks, particularly in scientific-literature QA and multi-hop reasoning, key evidence appears in *heterogeneous forms*. For example, in scientific-literature QA, extracting only a table or only a graph often fails because key evidence is distributed across **tabular measurements, citation chains, and narrative methodological descriptions**. In such settings, enforcing a *single* structure type (e.g., table-only or graph-only) leads to incomplete coverage and loss of reasoning signal. The following example serves as a compelling case study:
>
> ```
> Question: "Which filmmaker was known for animation, Lev Yilmaz or Pamela B. Green?"
>
> Document (Excerpt): Levni Yilmaz is an independent filmmaker, artist, and publisher best known for his "Tales of Mere Existence" animated comic series. The series began in 2002 and was shown at film festivals. Pamela B. Green is an award-winning American film director and producer known for her work in feature film titles and motion graphics. In 2018, Green's documentary "Be Natural" premiered at Cannes.
>
> Ground Truth: Levni Yilmaz
> ---
>
> 1. Extracted SSO (Table Only)
> {"type": "table", "content": "| Filmmaker | Roles | Known For |\n|---|---|---|\n| Lev Yilmaz | Independent Filmmaker, Artist, Publisher | Tales of Mere Existence |\n| Pamela B. Green | Film Director, Producer | Feature film titles, Motion graphics |" }
> Answer1:  Pamela B. Green is known for 'Motion graphics', which is related to animation. (wrong)
> ---
>
> 2. Extracted SSO (Hybrid Composition)
> [ {"type": "table",  "content": "| Filmmaker | Roles | Known For |\n|---|---|---|\n| Lev Yilmaz | Independent Filmmaker, Artist | Tales of Mere Existence |\n| Pamela B. Green | Film Director, Producer | Feature film titles |"  }, {"type": "supporting_text_span", "content": "Levni Yilmaz... best known for his 'Tales of Mere Existence'  animated comic series." },   {"type": "supporting_text_span",  "content": "Pamela B. Green... known for her work in feature film titles and motion graphics." } ]
> Answer 2: Lev Yilmaz was known for animation. (correct)
> ```
>
> **Hybrid and compositional SSO as a natural extension.** To address this, CoST is designed to be extensible toward hybrid SSO traces, where the output structure becomes a *composition* of table rows, graph nodes/edges, and supporting text spans. This richer structure space allows multiple structural forms to **coexist** within the same serialized output.
>
> **Structure analysis over an expanded structure space.** The structure-analysis module can be extended to select not only the *dominant* structure but also a composition of structures when a task requires them. This extension preserves the core CoST workflow: schema construction, alignment, and step-wise extraction, while enabling the system to handle cases where no single structure is adequate.
>
> In summary, while the current instantiation focuses on one SSO type per instance for compatibility with Loong’s evaluation protocol, CoST naturally supports hybrid structured outputs, and our ongoing experiments indicate that such compositional structures are valuable for complex, multi-evidence tasks.

---

> ### Author Response · Authors · 2025-11-23
>
> **3\. Explanation of lossy mapping \[W1\]**
>
> We agree that structured reasoning can introduce lossy mappings, especially for tasks that depend on non-aggregative or non-numeric cues. However, this does not limit CoST’s applicability. The **SSO trace is not confined to minimal tables or graphs**—its format can be enriched with provenance spans and short supporting text segments, allowing the structured output to retain the contextual nuances needed for subtler forms of reasoning. Moreover, as discussed in **W2**, **CoST naturally extends to hybrid structured-plus-text traces** in which structured elements (e.g., table rows or graph relations) are paired with semantically rich textual evidence. These hybrid representations substantially reduce information loss while preserving the benefits of schema-guided extraction. After **adopting this representation**, the updated results shown in the following table demonstrate improved performance on nuanced and semi-structured tasks, supporting the general applicability of the approach beyond numeric reasoning (Taking GPT-4o-mini and GPT-4o as examples).
>
> Table. Effectiveness of CoST augmented with textual evidence on the financial subset of Loong (SD indicates Structured Data, TE indicates Textual Evidence)
>
> | **Model** | **Spotlight Locating** |  | **Comparsion** |  | **Clustering** |  | **Chain of Reasoning** |  | **Overall** |  |
> | ----- | :---: | :---: | :---: | :---: | :---: | :---: | :---: | :---: | :---: | :---: |
> |  | **As** | **PR** | **As** | **PR** | **As** | **PR** | **As** | **PR** | **As** | **PR** |
> | GPt-4o-mini | 59.46 | 0.49 | 51.90 | 0.27 | 34.55 | 0.04 | **64.28** | **0.39** | 49.25 | 0.24 |
> | w /SD | 63.23 | 0.44 | 53.04 | 0.38 | 59.63 | 0.21 | 55.98 | 0.26 | 58.02 | 0.29 |
> | w / SD+TE | **67.56** | **0.49** | **53.24** | **0.39** | **61.94** | **0.26** | 63.75 | 0.38 | **61.61** | **0.35** |
> | GPT-4o | **73.95**  | **0.62**  | 50.50  | 0.28  | 44.29  | 0.09  | 57.05  | 0.28  | 53.47  | 0.26  |
> | w / SD | 62.11  | 0.33  | 63.27  | 0.41  | 68.06  | 0.29  | 53.32  | 0.22  | 62.51  | 0.30  |
> | w / SD+TE | 72.73 | 0.46 | **64.07** | **0.42** | **70.31** | **0.34** | **60.39** | **0.31** | **66.97** | **0.36** |

---

> ### Author Response · Authors · 2025-11-23
>
> **4\. Scale with task complexity \[W2, Q2\]**
>
> We thank the reviewer for raising this point. In response, we conducted additional analyses to model performance.
>
> **4.1 Scale with Table Size.** We analyzed how CoST+LLM performance scales with tabular complexity on the Finance subset of Loong benchmark, using the table cell count as the independent variable (x-axis).
>
> Table. LLM+CoST performance related to table complexity
>
>
> | **Model** | **Method** | **Table Complexity** |  |  |  |  |
> | :--- | :--- | :---: | :---: | :---: | :---: | :---: |
> |  |  | **0-10** | **11-20** | **21-30** | **31-40** | **41-** |
> | **Qwen2.5-14b-ins** | zero-shot | 83.53 | 76.71 | 76.12 | 75.21 | 73.18 |
> |  | LiteCoST (Ours) | 73.18 | 78.54 | 78.79 | 78.72 | 87.37 |
> | **GPT-4o** | zero-shot | 74.8 | 76.8 | 80.0 | 84.6 | 87.9 |
> |  | LiteCoST (Ours) | 76.6 | 81.3 | 83.1 | 85.6 | 88.2 |
>
> The table shows that CoST demonstrates robust performance scalability with respect to table size. Notably, CoST consistently maintains a **clear advantage over vanilla prompting** across the majority of table-size ranges. This performance gap becomes particularly pronounced in specific complexity splits, reaching a margin of **\+14.2 points** on the 41– split for Qwen2.5-14B-Instruct and **\+4.5 points** (11–20 split) for GPT-4o.
>
> **Result Analysis:** Performance improves monotonically with table size as larger tables offer richer schema constraints. Unlike small tables that lack sufficient structural cues, leading to open-ended inference errors, the increased information density in larger tables effectively guides the model, resulting in higher accuracy. Furthermore, the absence of a performance peak suggests that the tabular complexity in the current benchmark remains relatively simple for LLMs.
>
> **4.2 Scale with input length.** We also evaluated CoST and LiteCoST across different input-text lengths. Both models degrade gracefully as context grows, again outperforming baselines at every length range.
>
> Table. Performance (LLM score) versus document-set size for CoST with baseline methods
> | **Method** | **10K-50K** | **50K-100K** | **100K-200K** | **200K-250K** |
> | :--- | :---: | :---: | :---: | :---: |
> | LLM | 92.33 | 85.02 | 76.19 | 58.42 |
> | LLM+CoT | 87.14 | 84.96 | 79.62 | 60.49 |
> | LLM+CoST | **91.46** | **86.24** | **80.87** | **66.25** |
>
> Table. Performance (LLM score) versus document-set size for LiteCoST-tuned models with different base models.
> | **Model** | **10K-50K** | **50K-100K** | **100K-200K** | **200K-250K** |
> | :--- | :---: | :---: | :---: | :---: |
> | llama3.2-3b-ins | 47.28 | 52.61 | 48.67 | 45.11 |
> | qwen2-7b-ins | 72.52 | 67.61 | 58.08 | 47.07 |
> | llama3.1-8b-ins | 42.96 | 52.86 | 52.63 | 52.88 |
> | qwen2.5-14b-ins | 87.94 | 81.71 | 73.89 | 60.12 |
> | gpt-4o | 92.33 | 85.02 | 76.19 | 58.42 |
> | Qwen-LiteCoST | **89.35** | **83.37** | **78.29** | **64.89** |
>
> As shown in the above table (refer to **Figure 16 in Appendix C.4**), **CoST and LiteCoST both degrade gracefully as task complexity increases**, and their **improvements over baselines become more pronounced on harder splits**. For example, long-document inputs exhibit substantially larger gains than small/easy cases, indicating that structured reasoning is especially beneficial in high-complexity regimes.
>
> For graph-structured tasks, Loong contains **too few samples with high graph breadth/depth** to support a meaningful scaling analysis. The variance overwhelms any trend, so including a figure would be misleading. We will revisit this when richer graph data become available.

---

> ### Author Response · Authors · 2025-11-23
>
> **5\. Illustration of “Fuzzy” Reasoning \[Q1\]**
> We performed a targeted **failure mode analysis** on tasks requiring fuzzier, less structured reasoning. Across domains such as scientific QA, legal argumentation, and long-form narrative QA, we observe that **the text-chunk representation indeed acts as the fallback mechanism** when a strict table/graph structure is insufficient.
> Below we summarize the key failure modes and give concrete examples:
>
> * Failure Mode 1:  Loss of implicit discourse relations
>   **Case:** In scientific QA, the answer required identifying that *“Method A outperforms Method B only under low-resource conditions.”*
>   The table SSO captured numeric results but *missed the conditional narrative context*.
>   **Fallback outcome:** The chunk-based fallback preserved the surrounding sentences, enabling the model to recover the conditional statement, though with reduced precision.
> * ​​Failure Mode 2: Cross-sentence causal reasoning not expressible in a table or graph
>   **Case:** In narrative multi-hop QA, answering “Why did the policy ultimately fail?” required linking dispersed cues across paragraphs (public reaction → reduced funding → cancellation).
>   Neither table nor graph SSO expressed this soft causal chain reliably.
>   **Fallback outcome:** Chunk-based representation preserved the multi-sentence causal evidence, giving the model enough continuity to reconstruct the chain, although sometimes missing intermediate steps.
>
> In summary, under our task requirements, the chunk-based representation reliably acts as the fallback mechanism, preserving the necessary semantic context when a single structured form is insufficient.

---

> ### Author Response · Authors · 2025-11-23
>
> **6\. Suggestions about Application Generalization \[Q1\]**
> The CoST framework naturally generalizes to a broad range of applications because it offers a domain-agnostic recipe for constructing structured-reasoning supervision. Given any domain, one can (1) **define a lightweight schema**, (2) **generate CoST-style SSO traces** and final output using an LLM, and (3) **efficiently and effectively fine-tune** a smaller model for extraction and reasoning. This enables rapid adaptation to new domains without redesigning the core method.
>
> To further broaden applicability, several extensions are straightforward:
>
> * **Hybrid structured–text traces:** For applications where rigid tables/graphs cannot capture subtle discourse (e.g., legal reasoning, scientific claims), we can enrich CoST traces with **text chunks as supporting evidence**. This hybrid representation preserves semantic nuance while still offering structured anchors, enabling the system to handle fuzzier reasoning styles.
> * **CoST optimization:** CoST can be further optimized by **generating more fine-grained reasoning steps** and constructing a larger structured reasoning space (e.g., more intermediate nodes, sub-goals, or decomposition steps). A selection mechanism can then automatically choose the most useful steps for a given task or domain, enabling the model to adapt its reasoning granularity to application needs rather than relying on a single fixed pattern.
> * **Plug-and-Play Verifiers**：Introduce **Constraint-based Automatic Validation** that leverages constraints to improve distilled data quality.
>
> Constraint-based automatic validation provides an interpretable, schema-aware method for evaluating structured extraction quality by checking whether the extracted triples, tables, or graphs satisfy predefined logical and structural constraints. Instead of relying on human labels or downstream tasks, it verifies the internal consistency and plausibility of the extracted data.
>
> We are currently exploring the following types of constraints:
>
> * Domain and range constraints: enforce type correctness for relation subjects and objects.
> * Missing-value checks: detect incomplete or partially extracted records.
> * Functional-dependency checks: verify one-to-one or deterministic field relationships.
> * Schema-specific consistency rules: apply domain logic and inter-field coherence constraints.
>
> ---
>
> \[Concrete Example\]:
>
> Consider the task of extracting structured financial information, such as Accounts Payable, from a document. In the table below, the extracted output indicates a missing value (`N/A`) for *Acorda Therapeutics, Inc.*, while the ground truth in the raw document shows an actual amount of $9,932,000.
>
> | **Company** | **Accounts Payable** |
> | :--- | :---: |
> | Biostax Corp | $1,352,465 |
> | Acorda Therapeutics, Inc | N/A |
> | Exela Technologies, Inc. | $66,375 |
> | American Battery Materials, Inc. | $199,867 |
>
>
> By integrating constraint-based validation (*for example, requiring any monetary field mentioned in the source to appear as a non-empty, correctly formatted value in the extracted table*), we can automatically filter structurally inconsistent or semantically incomplete examples, thereby improving the quality and schema fidelity of the distilled data.
>
> ---
>
> **Overall**, these extensions (hybrid traces, richer reasoning spaces,  automated schema generation/selection, and ..) make the CoST framework more flexible and easier to deploy across a wide variety of applications.

---

> ### Author Response · Authors · 2025-11-23
>
> **7\. Analysis about the upper limit of methods \[Q2\]**
>
> Regarding the practical ceiling of the method, we have not observed a strict upper bound. Nevertheless, we have identified several factors that influence the practical upper limit of what CoST/LiteCoST can support:
>
> 1. **CoST quality**: The fidelity of the CoST traces, including correctness, completeness, and alignment, directly affects the quality of the resulting training data. High-quality CoST provides detailed, interpretable, and task-aligned supervision, significantly enhancing model generalization on complex IE tasks,  thus elevating the ceiling.
> 2. **Supervision volume**: More refinement iterations and more SSO traces directly expand the training signal, raising the ceiling; limited supervision naturally restricts the achievable performance.
> 3. **LLM \+CoST capacity:** In extreme cases, such as very large, multi-page tables or document sets with heavy cross-document dependencies, we begin to observe this ceiling, mostly due to base model limitations rather than CoST itself.
> 4. **Fine-tuned SLM capacity**: For LiteCoST, the upper bound is somehow defined by the teacher model. Since the distilled SLM inherits the reasoning behaviors demonstrated by the LLM+CoST teacher, its performance can closely approach but cannot surpass the teacher’s reasoning ability, establishing a natural ceiling for the fine-tuned compact model.
>
> These effects are also tied to the inherent structure and complexity of the underlying documents. Overall, our analysis suggests that CoST’s advantages become more pronounced with task complexity, and its practical ceiling is governed by supervision volume, CoST quality,  and model capacity.
>
> **8\. Presentation and Readability (Q3)**
>
> **8.1.** Figures 2, 3, and 4 have been updated to fix spelling errors (“Structure”, “Structuralizer”).
>
> **8.2.** In Section 1 (Introduction), “on-prem” is now written as “on-premises (on-prem)” with a brief clarification of meaning.
>
> **8.3.** The caption of Figure 5 now explicitly defines the axes (schema awareness, normalization, alignment, serialization, etc.). We also added a short clarifying sentence in Section 5.2.
>
> **8.4.** Figure 6 now reports both absolute values and improvement deltas using explicit “(+ΔX)” notation. The caption has been updated accordingly.
>
> **8.5.** Readability improvements for dense figures
>
> * Place the legend above the plot to enlarge the radar chart.
> * Increased font size, spacing, and labeling consistency.
> * Added clearer annotations and step numbers.
>
> These updates appear in Figures 3–6, improving readability while preserving content.
>
> 8.6. Clarifying CoST vs. LiteCoST analysis
>
> * In Section 5 (Experiments) and Section 6 (Analysis), we now consistently distinguish
>   CoST \+ LLM (off-the-shelf models using CoST traces) versus
>   LiteCoST (the distilled small model trained from CoST supervision).
> * We revised statements like “High-quality SSO improves LLM reasoning” to clarify that
>    the high-quality SSO is a result of CoST, not the base model itself.
> * All plots and tables now explicitly label the method categories (e.g., “LLM+CoST”, “LiteCoST”, and baseline LLMs).
>
> These clarifications improve interpretability for readers.

---

> ### Author Response · Authors · 2025-11-26
>
> Dear Reviewer 1HZK,
>
> We hope this message finds you well. As the discussion period is ongoing and **time is running short**, we wanted to ensure we have addressed all your concerns satisfactorily. If there are any additional points or feedback you'd like us to consider, please let us know. Your insights are invaluable to us, and we're eager to address any remaining issues to improve our work.
>
> Thank you for your time and effort in reviewing our paper.
>
> Best regards,
>
> Authors

---

### Author Response · Authors · 2025-12-01
**Summary of the Rebuttal Period (2/2)**

We are glad to see that the concerns raised by **Reviewer MkZ4** have been fully addressed. While it is unfortunate that we did not engage with the remaining reviewers, we understand that everyone has priorities, and the early closure of the discussion phase due to the OpenReview information leakage may have further limited the time available for additional feedback.

Nonetheless, we believe we have **comprehensively addressed all raised concerns** and incorporated the feedback into the revised version, based on the **elaborate experiments and clarifications** presented below. The key modifications and experiments made during the rebuttal period are summarized below:


### **Additional Experiments:**

> - **Expanded Domain Generalization:** We added evaluations on LongBench, including NarrativeQA, Qasper, HotpotQA, 2WikimQA, demonstrating **robust generalization to nuanced, less-structured domains,** and confirming that our approach yields substantial gains for LLMs while enabling SLMs to achieve performance comparable to proprietary large models. (Reviewer 1HZK W1; ufYT W3, NvTs W5; MkZ4 W1)
> - **Hybrid Trace Validation:** We validated hybrid structured-plus-text traces (enriching serialized structured outputs (SSOs) with provenance spans and textual evidence), demonstrating the performance gains achieved via this representation. (Reviewer 1HZK W1)
> - **Complexity Scaling Analysis:** We analyzed performance scaling against table complexity (cell count) and input length (up to 250k tokens), confirming that **our method maintains clear advantages over baselines in high-complexity regimes.** (Reviewer 1HZK W2,Q2)
> - **SLM Efficiency Benchmarking:** We compared our LiteCoST against fine-tuned baselines (ODIE, IEPIE, Struc-bench) as requested, demonstrating the **lowest latency while retaining superior accuracy** compared to prior methods. (Reviewer NvTs W4)
> - **Comparison with Prompt Optimization:** We compared our method against DSPy and standard CoT on LongBench, showing that **our Chain-of-Structured-Thought (CoST) outperforms automated prompt optimization frameworks.** (Reviewer NvTs W6)
> - **Structure Selection Reliability:** We conducted a human evaluation on structure type selection, confirming **high reliability and alignment** between model choices and human judgment without needing explicit correction. (Reviewer MkZ4 Q2)



### **Clarifications:**

> - **Highlighting Distinct Novelty:** We clarified our distinct contributions across three key dimensions and explicitly differentiated our work from prior "QA-by-structuring" systems. We have updated the "Introduction and Related Work" sections to incorporate these clarifications, highlighting **our novel, structure-first training pipeline that materially advances the ability of small models to perform reliable long-document reasoning.** (Reviewer ufYT W3; NvTs W1,W2,W8)
> - **Expanded Scope Analysis:** We complemented our experimental results with a thorough clarification of generalizability, showing why our method offers effective support for nuanced, non-rigid information. (Reviewer 1HZK W1)
> - **Failure Mode Analysis:** Beyond the additional experimental results, we presented a concrete case study illustrating scenarios that benefit from heterogeneous representations, and detailed how our "Hybrid Compositional SSO extension" addresses this by integrating structured data with narrative text spans. (Reviewer 1HZK Q1)
> - **Experimental Details:** We provided the citation for GPT-4o's parameter count and confirmed the use of identical prompt templates to ensure fair baseline comparison. (Reviewer ufYT W4; NvTs W3)
> - **Methodological Explanation:** We detailed the process reward computation, and justified our unit-weighted reward scheme as a standard configuration. (Reviewer ufYT W1; MkZ4 Q1)
> - **Cost-Efficiency Clarification:** We explicitly detailed the computational costs, showing that the training expense is **balanced out by efficient, low-cost SLM inference.** (Reviewer MkZ4 W2)
> - **Future Work Discussion:** We added some suggestions on how to make our method generalize to a broader set of applications such as introducing constraint-based automatic validation (Reviewer 1HZK Q1)
> - **General Paper Improvements:** We enhanced overall presentation quality by correcting typos, refining definitions, optimizing figure layouts (e.g., adding explicit deltas and clearer legends in Figure 5), and strictly distinguishing terminology between CoST and LiteCoST. (Reviewer 1HZK Q3)

In the revised manuscript, these updateds are temporarily highlighted in **blue** for your convenience to check.

---

We would like to express our deepest gratitude once again to the reviewers for their constructive feedback. We believe these extensive experiments and revisions have significantly strengthened our paper and thoroughly addressed the concerns raised, and we hope the final work will be of benefit to the ICLR community in this venue.

Best regards,

Authors

---

### Author Response · Authors · 2025-12-01
**Summary of the Rebuttal Period (1/2)**

Dear Reviewers and ACs,

As the discussion period is almost over, we would like to sincerely thank reviewers and ACs for their efforts in reviewing our paper and for the timely responses during the discussion period.


### **Summary**
First, we are encouraged that the reviewers consistently recognize the strengths of our work across novelty, methodology, experiments, and presentation:
> - **Novel and Well-Motivated:** The paper addresses the important and practical problem of “long-document QA” for both LLMs and SLMs, by introducing a novel and well-motivated framework that integrates a structure-first extraction pipeline with a two-stage SLM training strategy. Reviewers noted the idea is **"valuable" and "meaningful"**, and substantially improves the effectiveness of small models while preserving efficient inference, enabling reliable long-document reasoning. (Reviewer 1HZK, MkZ4)
> - **Methodologically Sound and Effective:** The integration of a structure-first extraction paradigm with a two-stage SLM training strategy is described as **sound, effective, and straightforward** (Reviewer 1HZK, NvTs), which is clearly demonstrated with strong in-domain evidence and some out-of-domain generalization. Robustness is further supported by extensive ablation studies and detailed methodological explanations. (Reviewer MkZ4)
> - **Experimentally Convincing:** The experiments are **solid, comprehensive, and convincing** (Reviewer NvTS, MkZ4), with LiteCoST outperfoming strong baselines, including standard LLMs and other fine-tuned models, on popluar Loong benchmark across multiple subtasks. (Reviewer 1HZK, ufYT, NvTS, MkZ4)
> - **Clearly Presented:** The paper is **well-written and easy to follow**. (Reviewer NvTS, MkZ4)

---

### Meta-Review · Area_Chair_uWG6 · 2026-01-03

**Summary:**

Reviewers expressed concerns over the method's limited novelty as an integration of existing techniques like structured prompting and SFT+GRPO without sufficient unique contributions, narrow evaluation scope primarily on finance/legal domains lacking generalization to nuanced or scientific QA, insufficient ablations on reward sensitivity and task complexity scaling, unclear prompting details and comparisons to baselines like DSPy or StructRAG, and reliance on costly LLM supervision for traces/rewards, leading to a suggested BORDERLINE paper given strong empirical results but incremental advances.

**Reviewer Concerns:**

The rebuttal addressed domain generalization via LongBench evaluations, hybrid trace validation for composite structures, complexity scaling analysis, SLM latency benchmarking against baselines, comparisons to prompt optimization like DSPy, and structure selection reliability through human evaluation. Outstanding concerns include full GRPO reward weighting sensitivity across domains, comprehensive total compute costs beyond high-level estimates, and deeper differentiation from prior QA-by-structuring systems despite added discussions.

**Reviewer Scores:**

Reviewer 1HZK (initial 6) would likely increase the score, appreciating added LongBench generalization, hybrid traces, and complexity analyses. Reviewer ufYT (initial 4) might remain at 4, with novelty and process reward details partially met but benchmark scope still limited. Reviewer NvTs (initial 4) could rise to 6, as rebuttal included prompt clarifications, DSPy comparisons, and cross-domain results; Reviewer MkZ4 (initial 6) would maintain 6, with cost-efficiency and domain concerns resolved.

---

### Decision · Program_Chairs · 2026-01-26

Accept (Poster)